# HETEROSCEDASTIC TEMPORAL VARIATIONAL AUTOENCODER FOR IRREGULAR TIME SERIES

**Satya Narayan Shukla & Benjamin M. Marlin**
College of Information and Computer Sciences
University of Massachusetts Amherst
Amherst, MA 01003, USA
`{snshukla,marlin}@cs.umass.edu`

## ABSTRACT

Irregularly sampled time series commonly occur in several domains where they present a significant challenge to standard deep learning models. In this paper, we propose a new deep learning framework for probabilistic interpolation of irregularly sampled time series that we call the *Heteroscedastic Temporal Variational Autoencoder* (HeTVAE). HeTVAE includes a novel input layer to encode information about input observation sparsity, a temporal VAE architecture to propagate uncertainty due to input sparsity, and a heteroscedastic output layer to enable variable uncertainty in output interpolations. Our results show that the proposed architecture is better able to reflect variable uncertainty through time due to sparse and irregular sampling than a range of baseline and traditional models, as well as recent deep latent variable models that use homoscedastic output layers.[1]

## 1 INTRODUCTION

In this paper, we propose a novel deep learning framework for probabilistic interpolation of irregularly sampled time series. Irregularly sampled time series data occur in multiple scientific and industrial domains including finance (Manimaran et al., 2006), climate science (Schulz & Stattegger, 1997) and healthcare (Marlin et al., 2012; Yadav et al., 2018). In some domains including electronic health records and mobile health studies (Cheng et al., 2017), there can be significant variation in inter-observation intervals through time. This is due to the complexity of the underlying observation processes that can include "normal" variation in observation times combined with extended, block-structured periods of missingness. For example, in the case of ICU EHR data, this can occur due to patients being moved between different locations for procedures or tests, resulting in missing physiological sensor data for extended periods of time. In mobile health studies, the same problem can occur due to mobile sensor batteries running out, or participants forgetting to wear or carry devices.

In such situations, it is of critical importance for interpolation models to be able to correctly reflect the variable input uncertainty that results from variable observation sparsity so as not to provide overly confident inferences. However, modeling time series data subject to irregular sampling poses a significant challenge to machine learning models that assume fully-observed, fixed-size feature representations (Marlin et al., 2012; Yadav et al., 2018; Shukla & Marlin, 2021b). The main challenges in dealing with such data include the presence of variable time gaps between the observation time points, partially observed feature vectors caused by the lack of temporal alignment across different dimensions, as well as different data cases, and variable numbers of observations across dimensions and data cases. Significant recent work has focused on developing specialized models and architectures to address these challenges in modeling irregularly sampled multivariate time series (Li & Marlin, 2015; 2016; Lipton et al., 2016; Futoma et al., 2017; Che et al., 2018a; Shukla & Marlin, 2019; Rubanova et al., 2019; Horn et al., 2020; Li & Marlin, 2020; Shukla & Marlin, 2021a; De Brouwer et al., 2019; Tan et al., 2020; Kidger et al., 2020).

Recently, Shukla & Marlin (2021a) introduced the Multi-Time Attention Network (mTAN) model, a variational autoencoder (VAE) architecture for continuous-time interpolation of irregularly sampled

---

[1]Implementation available at `https://github.com/reml-lab/hetvae`

time series. This model was shown to provide state-of-the-art classification and deterministic interpolation performance. However, like many VAEs, the mTAN architecture produces a homoscedastic output distribution conditioned on the latent state. This means that the model can only reflect uncertainty due to variable input sparsity through variations in the VAE latent state. As we will show, this mechanism is insufficient to capture differences in uncertainty over time. On the other hand, Gaussian Process Regression-based (GPR) methods (Rasmussen & Williams, 2006) have the ability to reflect variable uncertainty through the posterior inference process. The main drawbacks of GPR-based methods are their significantly higher run times during both training and inference, and the added restriction to define positive definite covariance functions for multivariate time series.

In this work, we propose a novel encoder-decoder architecture for multivariate probabilistic time series interpolation that we refer to as the *Heteroscedastic Temporal Variational Autoencoder* or HeTVAE. HeTVAE aims to address the challenges described above by encoding information about input sparsity using an uncertainty-aware multi-time attention network (UnTAN), flexibly capturing relationships between dimensions and time points using both probabilistic and deterministic latent pathways, and directly representing variable output uncertainty via a heteroscedastic output layer.

The proposed UnTAN layer generalizes the previously introduced mTAN layer with an additional intensity network that can more directly encode information about input uncertainty due to variable sparsity. The proposed UnTAN layer uses an attention mechanism to produce a distributed latent representation of irregularly sampled time series at a set of reference time points. The UnTAN module thus provides an interface between input multivariate, sparse and irregularly sampled time series data and more traditional deep learning components that expect fixed-dimensional or regularly spaced inputs. We combat the presence of additional local optima that arises from the use of a heteroscedastic output layer by leveraging an augmented training objective where we combine the ELBO loss with an uncertainty agnostic loss component. The uncertainty agnostic component helps to prevent learning from converging to local optima where the structure in data is explained as noise.

We evaluate the proposed architecture on both synthetic and real data sets. Our approach outperforms a variety of baseline models and recent approaches in terms of log likelihood, which is our primary metric of interest in the case of probabilistic interpolation. Finally, we perform ablation testing of different components of the architecture to assess their impact on interpolation performance.

## 2    RELATED WORK

Keeping in mind the focus of this work, we concentrate our discussion of related work on deterministic and probabilistic approaches applicable to the interpolation and imputation tasks.

**Deterministic Interpolation Methods:** Deterministic interpolation methods can be divided into filtering and smoothing-based approaches. Filtering-based approaches infer the values at a given time by conditioning only on past observations. For example, Han-Gyu Kim et al. (2017) use a unidirectional RNN for missing data imputation that conditions only on data from the relative past of the missing observations. On the other hand, smoothing-based methods condition on all possible observations (past and future) to infer any unobserved value. For example, Yoon et al. (2018) and Cao et al. (2018) present missing data imputation approach based on multi-directional and bi-directional RNNs. These models typically use the gated recurrent unit with decay (GRU-D) model (Che et al., 2018a) as a base architecture for dealing with irregular sampling. Interpolation-prediction networks take a different approach to interfacing with irregularly sampled data that is based on the use of temporal kernel smoother-based layers (Shukla & Marlin, 2019). Shan & Oliva (2021) propose hierarchical imputation strategy based on set-based architectures for imputation in irregularly sampled time series. Of course, the major disadvantage of deterministic interpolation approaches is that they do not express uncertainty over output interpolations and thus can not be applied to the problem of probabilistic interpolation without modifications.

**Probabilistic Interpolation Methods:** The two primary building blocks for probabilistic interpolation and imputation of multivariate irregularly sampled time series are Gaussian processes regression (GPR) (Rasmussen & Williams, 2006) and variational autoencoders (VAEs) (Rezende et al., 2014; Kingma & Welling, 2014). GPR models have the advantage of providing an analytically tractable full joint posterior distribution over interpolation outputs when conditioned on irregularly sampled input data. Commonly used covariance functions have the ability to translate variable input obser-

vation density into variable interpolation uncertainty. GPR-based models have been used as the core of several approaches for supervised learning and forecasting with irregularly sampled data (Ghassemi et al., 2015; Li & Marlin, 2015; 2016; Futoma et al., 2017). However, GPR-based models can become somewhat cumbersome in the multivariate setting due to the positive definiteness constraint on the covariance function (Rasmussen & Williams, 2006). The use of separable covariance functions is one common approach to the construction of GPR models over multiple dimensions (Bonilla et al., 2008), but this construction requires all dimensions to share the same temporal kernel parameters. A further drawback of GP-based methods is their significantly higher run times relative to deep learning-based models when applied to larger-scale data (Shukla & Marlin, 2019).

Variational autoencoders (VAEs) combine probabilistic latent states with deterministic encoder and decoder networks to define a flexible and computationally efficient class of probabilistic models that generalize classical factor analysis (Kingma & Welling, 2014; Rezende et al., 2014). Recent research has seen the proposal of several new VAE-based models for irregularly sampled time series. Chen et al. (2018) proposed a latent ordinary differential equation (ODE) model for continuous-time data using an RNN encoder and a neural ODE decoder. Building on the prior work of Chen et al. (2018), Rubanova et al. (2019) proposed a latent ODE model that replaces the RNN with an ODE-RNN model as the encoder. Li et al. (2020) replace the deterministic ODEs with stochastic differential equations(SDEs). Norcliffe et al. (2021) extends the prior work on neural ode by combining it with neural processes (Garnelo et al., 2018). Shukla & Marlin (2021a) proposed the Multi-Time Attention Network (mTAN) model, a VAE-based architecture that uses a multi-head temporal cross attention encoder and decoder module (the mTAND module) to provide the interface to multivariate irregularly sampled time series data. Fortuin et al. (2020) proposed a VAE-based approach for the task of smoothing in multivariate time series with a Gaussian process prior in the latent space to capture temporal dynamics. Garnelo et al. (2018); Kim et al. (2019) used heteroscedastic output layers to represent uncertainty in case of fixed dimensional inputs but these approaches are not applicable to irregularly sampled time series.

Similar to the mTAN model, the Heteroscedastic Temporal Variational Autoencoder (HeTVAE) model proposed in this work is an attention-based VAE architecture. The primary differences are that mTAN uses a homoscedastic output distribution that assumes constant uncertainty and that the mTAN model's cross attention operation normalizes away information about input sparsity. These limitations are problematic in cases where there is variable input density through time resulting in the need for encoding, propagating, and reflecting that uncertainty in the output distribution. As we describe in the next section, HeTVAE addresses these issues by combining a novel sparsity-sensitive encoder module with a heteroscedastic output distribution and parallel probabilistic and deterministic pathways for propagating information through the model. Another important difference relative to these previous methods is that HeTVAE uses an augmented learning objective to address the underfitting of predictive variance caused by the use of the heteroscedastic layer.

## 3 PROBABILISTIC INTERPOLATION WITH THE HETVAE

In this section, we present the proposed architecture for probabilistic interpolation of irregularly sampled time series, the Heteroscedastic Temporal Variational Autoencoder (HeTVAE). HeTVAE leverages a sparsity-aware layer as the encoder and decoder in order to represent input uncertainty and propagate it to output interpolations. We begin by introducing notation. We then describe the architecture of the encoder/decoder network followed by the complete HeTVAE architecture.

### 3.1 NOTATION

We let $\mathcal{D} = \{\mathbf{s}_n | n = 1, ..., N\}$ represent a data set containing $N$ data cases. An individual data case consists of a $D$-dimensional, sparse and irregularly sampled multivariate time series $\mathbf{s}_n$. Different dimensions $d$ of the multivariate time series can have observations at different times, as well as different total numbers of observations $L_{dn}$. We follow the series-based representation of irregularly sampled time series (Shukla & Marlin, 2021b) and represent time series $d$ for data case $n$ as a tuple $\mathbf{s}_{dn} = (\mathbf{t}_{dn}, \mathbf{x}_{dn})$ where $\mathbf{t}_{dn} = [t_{1dn}, ..., t_{L_{dn}dn}]$ is the list of time points at which observations are defined and $\mathbf{x}_{dn} = [x_{1dn}, ..., x_{L_{dn}dn}]$ is the corresponding list of observed values. We drop the data case index $n$ for brevity when the context is clear.

## 3.2 REPRESENTING INPUT SPARSITY

As noted in the previous section, the mTAN encoder module does not represent information about input sparsity due to the normalization of the attention weights. To address this issue, we propose an augmented module that we refer to as an **Un**certainty Aware Multi-**T**ime **A**ttention **N**etwork (UnTAN). The UnTAN module is shown in Figure 1a. This module includes two encoding pathways that leverage a shared time embedding function and a shared attention function. The first encoding pathway (the intensity pathway, INT) focuses on representing information about the sparsity of observations while the second encoding pathway (the value pathway, VAL) focuses on representing information about values of observations. The outputs of these two pathways are concatenated and mixed via a linear layer to define the final output of the module. The mathematical description of the module is given in Equations 1 to 3 and is explained in detail below.

$$\text{int}_h(r_k, \mathbf{t}_d) = \frac{\texttt{pool}(\{\exp(\alpha_h(r_k, t_{id})) \mid t_{id} \in \mathbf{t}_d\})}{\texttt{pool}(\{\exp(\alpha_h(r_k, t_{i'u})) \mid t_{i'u} \in \mathbf{t}_u\})} \tag{1}$$

$$\text{val}_h(r_k, \mathbf{t}_d, \mathbf{x}_d) = \frac{\texttt{pool}(\{\exp(\alpha_h(r_k, t_{id})) \cdot x_{id} \mid t_{id} \in \mathbf{t}_d, x_{id} \in \mathbf{x}_d\})}{\texttt{pool}(\{\exp(\alpha_h(r_k, t_{i'd})) \mid t_{i'd} \in \mathbf{t}_d\})} \tag{2}$$

$$\alpha_h(t, t') = \left( \frac{\phi_h(t)\mathbf{w}\mathbf{v}^T \phi_h(t')^T}{\sqrt{d_e}} \right) \tag{3}$$

**Time Embeddings and Attention Weights:** Similar to the mTAN module, the UnTAN module uses time embedding functions $\phi_h(t)$ to project univariate time values into a higher dimensional space. Each time embedding function is a one-layer fully connected network with a sine function non-linearity $\phi_h(t) = \sin(\omega \cdot t + \beta)$. We learn $H$ time embeddings each of dimension $d_e$. $\mathbf{w}$ and $\mathbf{v}$ are the parameters of the scaled dot product attention function $\alpha_h(t, t')$ shown in Equation 3. The scaling factor $1/\sqrt{d_e}$ is used to normalize the dot product to counteract the growth in the dot product magnitude with increase in the time embedding dimension $d_e$.

**Intensity Encoding:** The intensity encoding pathway is defined by the function $\text{int}_h(r_k, \mathbf{t}_d)$ shown in Equation 1. The inputs to the intensity function are a query time point $r_k$ and a vector $\mathbf{t}_d$ containing all the time points at which observations are available for dimension $d$. The numerator of the intensity function exponentiates the attention weights between $r_k$ and each time point in $\mathbf{t}_d$ to ensure positivity, then pools over the observed time points. The denominator of this computation is identical to the numerator, but the set of time points $\mathbf{t}_u$ that is pooled over is the union over all observed time points for dimension $d$ from all data cases.

Intuitively, if the largest attention weight between $r_k$ and any element of $\mathbf{t}_d$ is small relative to attention weights between $r_k$ and the time points in $\mathbf{t}_u$, then the output of the intensity function will be low. Importantly, due to the use of the non-linear time embedding function, pairs of time points with high attention weights do not necessarily have to be close together in time meaning the notion of intensity that the network expresses is significantly generalized.

We also note that different sets could be used for $\mathbf{t}_u$ including a regularly spaced set of reference time points. One advantage of using the union of all observed time points is that it fixes the maximum value of the intensity function at 1. The two pooling functions applicable in the computation of the intensity function are $max$ and $sum$. If the time series is sparse, $max$ works well because using $sum$ in the sparse case can lead to very low output values. In a more densely observed time series, either $sum$ or $max$ can be used.

**Value Encoding:** The value encoding function $\text{val}_h(r_k, \mathbf{t}_d, \mathbf{x}_d)$ is presented in Equation 2 in a form that highlights the symmetry with the intensity encoding function. The primary differences are that $\text{val}_h(r_k, \mathbf{t}_d, \mathbf{x}_d)$ takes as input both observed time points $\mathbf{t}_d$ and their corresponding values $\mathbf{x}_d$, and the denominator of the function pools over $\mathbf{t}_d$ itself. While different pooling options could be used for this function, in practice we use sum-based pooling. These choices lead to a function $\text{val}_h(r_k, \mathbf{t}_d, \mathbf{x}_d)$ that interpolates the observed values at the query time points using softmax weights derived from the attention function. The values of observed points with higher attention weights contribute more to the output value. This structure is equivalent to that used in the mTAN module when sum-based pooling is used. We can also clearly see that this function on its own can not represent

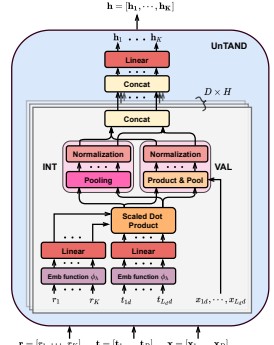 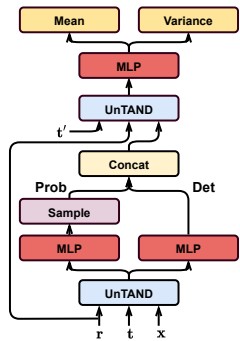

(a) Uncertainty Aware Multi-Time Attention Networks    (b) Heteroscedastic Temporal VAE

Figure 1: (a) Architecture of UnTAN module. This module takes D-dimensional irregularly sampled time points $\mathbf{t} = [\mathbf{t}_1, \cdots, \mathbf{t}_D]$ and corresponding observations $\mathbf{x} = [\mathbf{x}_1, \cdots, \mathbf{x}_D]$ as keys and values and produces a fixed dimensional representation at the query time points $\mathbf{r} = [r_1, \cdots, r_K]$. Shared time embedding and attention function provide input to parallel intensity (INT) and value (VAL) encoding networks, whose outputs are subsequently fused via concatenation and an additional linear encoding layer. (b) Architecture of HeTVAE consisting of the UnTAND module to represent input uncertainty, parallel probabilistic (Prob) and deterministic (Det) encoding paths, and a heteroscedastic output layer that aims to reflect uncertainty due to input sparsity in the output distribution.

information about input sparsity due to the normalization over $\mathbf{t}_d$. Indeed, the function is completely invariant to an additive decrease in all of the attention weights $\alpha'_h(r_k, t_{id}) = \alpha_h(r_k, t_{id}) - \delta$.

**Module Output:**   The last stage of the UnTAN module concatenates the value and intensity pathway representations and then linearly weights them together to form the final J-dimensional representation that is output by the module. The parameters of this linear stage of the model are $U_{hdj}^{int}$ and $U_{hdj}^{val}$. The value of the $j^{\text{th}}$ dimension of the output at a query time point $r_k$ is given by Equation 4.

$$\text{UnTAN}(r_k, \mathbf{t}, \mathbf{x})[j] = \sum_{h=1}^{H} \sum_{d=1}^{D} \begin{bmatrix} \text{int}_h(r_k, \mathbf{t}_d) \\ \text{val}_h(r_k, \mathbf{t}_d, \mathbf{x}_d) \end{bmatrix}^T \begin{bmatrix} U_{hdj}^{int} \\ U_{hdj}^{val} \end{bmatrix} \tag{4}$$

Finally, we note that the UnTAN module defines a continuous function of $t$ given an input time series and hence cannot be directly incorporated into standard neural network architectures. We adapt the UnTAN module to produce fully observed fixed-dimensional discrete sequences by materializing its output at a set of reference time points. Reference time points can be fixed set of regularly spaced time points or may need to depend on the input time series. For a given set of reference time points $\mathbf{r} = [r_1, \cdots, r_K]$, the discretized UnTAN module UnTAND$(\mathbf{r}, \mathbf{t}, \mathbf{x})$ is defined as UnTAND$(\mathbf{r}, \mathbf{t}, \mathbf{x})[i] = \text{UnTAN}(r_i, \mathbf{t}, \mathbf{x})$. This module takes as input the time series $\mathbf{s} = (\mathbf{t}, \mathbf{x})$ and the set of reference time points $\mathbf{r}$ and outputs a sequence of $K$ UnTAN embeddings, each of dimension $J$ corresponding to each reference point. As described in the next section, we use the UnTAND module to provide an interface between sparse and irregularly sampled data and fully connected MLP network structures.

### 3.3   THE HETVAE MODEL

In this section, we describe the overall architecture of the HeTVAE model, as shown in Figure 1b.

**Model Architecture:**   The HeTVAE consists of parallel deterministic and probabilistic pathways for propagating input information to the output distribution, including information about input sparsity. We begin by mapping the input time series $\mathbf{s} = (\mathbf{t}, \mathbf{x})$ through the UnTAND module along with a collection of $K$ reference time points $\mathbf{r}$. In the probabilistic path, we construct a distribution over latent variables at each reference time point using a diagonal Gaussian distribution $q$ with mean and variance output by fully connected layers applied to the UnTAND output embeddings

$\mathbf{h}^{enc} = [\mathbf{h}_1^{enc}, \cdots, \mathbf{h}_K^{enc}]$ as shown in Equation 6. In the deterministic path, the UnTAND output embeddings $\mathbf{h}^{enc}$ are passed through a feed-forward network $g$ to produce a deterministic temporal representation (at each reference point) of the same dimension as the probabilistic latent state.

The decoder takes as input the representation from both pathways along with the reference time points and a set of query points $\mathbf{t}'$ (Eq 8). The UnTAND module produces a sequence of embeddings $\mathbf{h}^{dec} = [\mathbf{h}_1^{dec}, \cdots, \mathbf{h}_{|\mathbf{t}'|}^{dec}]$ corresponding to each time point in $\mathbf{t}'$. The UnTAND embeddings are then independently decoded using a fully connected decoder $f^{dec}$ and the result is used to parameterize the output distribution. We use a diagonal covariance Gaussian distribution where both the mean $\boldsymbol{\mu} = [\boldsymbol{\mu}_1, \cdots, \boldsymbol{\mu}_{|\mathbf{t}'|}], \boldsymbol{\mu}_i \in \mathbb{R}^D$ and variance $\boldsymbol{\sigma}^2 = [\boldsymbol{\sigma}_1^2, \cdots, \boldsymbol{\sigma}_{|\mathbf{t}'|}^2], \boldsymbol{\sigma}_i^2 \in \mathbb{R}^D$ are predicted for each time point by the final decoded representation as shown in Eq 9. The generated time series is sampled from this distribution and is given by $\hat{\mathbf{s}} = (\mathbf{t}', \mathbf{x}')$ with all data dimensions observed.

The complete model is described below. We define $q_\gamma(\mathbf{z}|\mathbf{r}, \mathbf{s})$ to be the distribution over the probabilistic latent variables $\mathbf{z} = [\mathbf{z}_1, \cdots, \mathbf{z}_K]$ induced by the input time series $\mathbf{s} = (\mathbf{t}, \mathbf{x})$ at the reference time points $\mathbf{r}$. We define the prior $p(\mathbf{z}_i)$ over the latent states to be a standard multivariate normal distribution. We let $p_\theta^{het}(x'_{id} | \mathbf{z}^{cat}, t'_{id})$ define the final probability distribution over the value of time point $t'_{id}$ on dimension $d$ given the concatenated latent state $\mathbf{z}^{cat} = [\mathbf{z}_1^{cat}, \cdots, \mathbf{z}_K^{cat}]$. $\gamma$ and $\theta$ represent the parameters of all components of the encoder and decoder respectively.

$$\mathbf{h}^{enc} = \text{UnTAND}^{enc}(\mathbf{r}, \mathbf{t}, \mathbf{x}) \tag{5}$$

$$\mathbf{z}_k \sim q_\gamma(\mathbf{z}_k \,|\, \boldsymbol{\mu}_k, \boldsymbol{\sigma}_k^2), \quad \boldsymbol{\mu}_k = f_\mu^{enc}(\mathbf{h}_k^{enc}), \quad \boldsymbol{\sigma}_k^2 = f_\sigma^{enc}(\mathbf{h}_k^{enc}) \tag{6}$$

$$\mathbf{z}_k^{cat} = \text{concat}(\mathbf{z}_k, g(\mathbf{h}_k^{enc})) \tag{7}$$

$$\mathbf{h}^{dec} = \text{UnTAND}^{dec}(\mathbf{t}', \mathbf{r}, \mathbf{z}^{cat}) \tag{8}$$

$$p_\theta^{het}(x'_{id} \,|\, \mathbf{z}^{cat}, t'_{id}) = \mathcal{N}(x'_{id}; \boldsymbol{\mu}_i[d], \boldsymbol{\sigma}_i^2[d]), \quad \boldsymbol{\mu}_i = f_\mu^{dec}(\mathbf{h}_i^{dec}), \quad \boldsymbol{\sigma}_i^2 = f_\sigma^{dec}(\mathbf{h}_i^{dec}) \tag{9}$$

$$x'_{id} \sim p_\theta^{het}(x'_{id} \,|\, \mathbf{z}^{cat}, t'_{id}) \tag{10}$$

Compared to the constant output variance used to train the mTAN-based VAE model proposed in prior work (Shukla & Marlin, 2021a), our proposed model produces a heteroscedastic output distribution that we will show provides improved modeling for the probabilistic interpolation task. However, the increased complexity of the model's output representation results in an increased space of local optima. We address this issue using an augmented learning objective, as described in the next section. Finally, we note that we can easily obtain a simplified homoscedastic version of the model with constant output variance $\sigma_c^2$ using the alternate final output distribution $p_\theta^c(x'_{id} \,|\, \mathbf{z}, t'_{id}) = \mathcal{N}(x'_{id}; \boldsymbol{\mu}_i[d], \sigma_c^2)$.

**Augmented Learning Objective:** To learn the parameters of the HeTVAE framework given a data set of sparse and irregularly sampled time series, we propose an augmented learning objective based on a normalized version of the evidence lower bound (ELBO) combined with an uncertainty agnostic scaled squared loss. We normalize the contribution from each data case by the total number of observations so that the effective weight of each data case in the objective function is independent of the total number of observed values. The augmented learning objective is defined below. $\boldsymbol{\mu}_n$ is the predicted mean over the test time points as defined in Equation 9. Also recall that the concatenated latent state $\mathbf{z}^{cat}$ depends directly on the probabilistic latent state $\mathbf{z}$.

$$\mathcal{L}_{\text{NVAE}}(\theta, \gamma) = \sum_{n=1}^{N} \frac{1}{\sum_d L_{dn}} \Big( \mathbb{E}_{q_\gamma(\mathbf{z}|\mathbf{r}, \mathbf{s}_n)}[\log p_\theta^{het}(\mathbf{x}_n | \mathbf{z}_n^{cat}, \mathbf{t}_n)] - D_{\text{KL}}(q_\gamma(\mathbf{z}|\mathbf{r}, \mathbf{s}_n) || p(\mathbf{z})) \tag{11}$$

$$- \lambda \, \mathbb{E}_{q_\gamma(\mathbf{z}|\mathbf{r}, \mathbf{s}_n)} \|\mathbf{x}_n - \boldsymbol{\mu}_n\|_2^2 \Big)$$

$$D_{\text{KL}}(q_\gamma(\mathbf{z}|\mathbf{r}, \mathbf{s}_n) || p(\mathbf{z})) = \sum_{i=1}^{K} D_{\text{KL}}(q_\gamma(\mathbf{z}_i|\mathbf{r}, \mathbf{s}_n) || p(\mathbf{z}_i)) \tag{12}$$

$$\log p_\theta^{het}(\mathbf{x}_n | \mathbf{z}_n^{cat}, \mathbf{t}_n) = \sum_{d=1}^{D} \sum_{j=1}^{L_{dn}} \log p_\theta^{het}(x_{jdn} | \mathbf{z}_n^{cat}, t_{jdn}) \tag{13}$$

We include the uncertainty agnostic scaled squared loss term to counteract the propensity of the heteroscedastic model to become stuck in poor local optima where the mean is essentially flat and

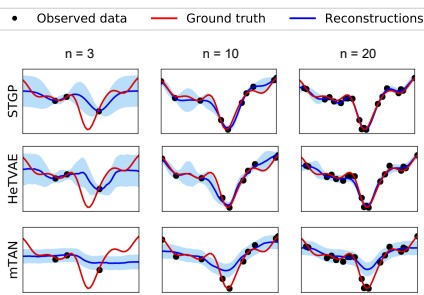

Figure 2: We show example interpolations on the synthetic dataset. The set of 3 columns correspond to interpolation results with increasing numbers of observed points: 3, 10 and 20 respectively. The first, second and third rows correspond to STGP, HeTVAE and HTVAE mTAN respectively. The shaded region corresponds to ± one standard deviation. STGP and HetVAE exhibit variable output uncertainty in response to input sparsity while mTAN does not.

all of the structure in the data is explained as noise. This happens because the model has the ability to learn larger variances at the output, which allows the mean to underfit the data. The extra component (scaled squared loss) helps to push the optimization process to find more informative parameters by introducing a fixed penalty for the mean deviating from the data. As we will show in the experiments, the use of this augmented training procedure has a strong positive impact on final model performance. Since, we are focusing on the interpolation task, we train the HeTVAE by maximizing the augmented learning objective (Equation 11) on the interpolated time points (more details on training has been provided in the experimental protocols in Section 4).

## 4 EXPERIMENTS

In this section, we present interpolation experiments using a range of models on three real-world data sets. PhysioNet Challenge 2012 (Silva et al., 2012) and MIMIC-III (Johnson et al., 2016) consist of multivariate, sparse and irregularly sampled time series data. We also perform experiments on the Climate dataset (Menne et al., 2016), consisting of multi-rate time series. We also show qualitative results on a synthetic dataset. Details of each dataset can be found in the Appendix A.6.1.

**Experimental Protocols:**  We randomly divide the real data sets into a training set containing 80% of the instances, and a test set containing the remaining 20% of instances. We use 20% of the training data for validation. In the interpolation task, we condition on a subset of available points and produce distributions over the rest of the time points. On the real-world datasets, we perform interpolation experiments by conditioning on $50\%$ of the available points. At test time, the values of observed points are conditioned on and each model is used to infer single time point marginal distributions over values at the rest of the available time points in the test instance. In the case of methods that do not produce probabilistic outputs, we make mean predictions. In the case of the synthetic dataset where we have access to all true values, we use the observed points to infer the values at the rest of the available points. We repeat each real data experiment five times using different random seeds to initialize the model parameters. We assess performance using the negative log likelihood, which is our primary metric of interest. We also report mean squared and mean absolute error. For all experiments, we select hyper-parameters on the held-out validation set using grid search and then apply the best trained model to the test set. The hyper-parameter ranges searched for each model and dataset are fully described in Appendix A.5.

**Models:**  We compare our proposed model HeTVAE to several probabilistic and deterministic interpolation methods. We compare to two Gaussian processes regression (GPR) approaches. The most basic GP model for multivariate time series fits one GPR model per dimension. This approach is known as a single task GP model (**STGP**) (Rasmussen & Williams, 2006). A potentially better option is to model data using a Multi Task GP (**MTGP**) (Bonilla et al., 2008). This approach models the correlations both across different dimensions and across time by defining a kernel expressed as the Hadamard product of a temporal kernel (as used in the STGP) and a task kernel. We also compare to several VAE-based approaches. These approaches use a homoscedastic output distribution with different encoder and decoder architectures. **HVAE RNN** employs a gated recurrent unit network (Chung et al., 2014) as encoder and decoder, **HVAE RNN-ODE** (Chen et al., 2018) replaces the RNN decoder with a neural ODE, **HVAE ODE-RNN-ODE** (Rubanova et al., 2019) employs

Table 1: Interpolation performance on PhysioNet.

| Model | Negative Log Likelihood | Mean Absolute Error | Mean Squared Error |
|---|---|---|---|
| Mean Imputation | − | $0.7396 \pm 0.0000$ | $1.1634 \pm 0.0000$ |
| Forward Imputation | − | $0.4840 \pm 0.0000$ | $0.9675 \pm 0.0000$ |
| Single-Task GP | $0.7875 \pm 0.0005$ | $0.4075 \pm 0.0001$ | $0.6110 \pm 0.0003$ |
| Multi-Task GP | $0.9250 \pm 0.0040$ | $0.4178 \pm 0.0007$ | $0.7381 \pm 0.0051$ |
| HVAE RNN | $1.5220 \pm 0.0019$ | $0.7634 \pm 0.0014$ | $1.2061 \pm 0.0038$ |
| HVAE RNN-ODE | $1.4946 \pm 0.0025$ | $0.7372 \pm 0.0026$ | $1.1545 \pm 0.0058$ |
| HVAE ODE-RNN-ODE | $1.2906 \pm 0.0019$ | $0.5334 \pm 0.0020$ | $0.7622 \pm 0.0027$ |
| HTVAE mTAN | $1.2426 \pm 0.0028$ | $0.5056 \pm 0.0004$ | $0.7167 \pm 0.0016$ |
| **HeTVAE** | $\mathbf{0.5542 \pm 0.0209}$ | $\mathbf{0.3911 \pm 0.0004}$ | $\mathbf{0.5778 \pm 0.0020}$ |

Table 2: Interpolation performance on MIMIC-III.

| Model | Negative Log Likelihood | Mean Absolute Error | Mean Squared Error |
|---|---|---|---|
| Mean Imputation | − | $0.7507 \pm 0.0000$ | $0.9842 \pm 0.0000$ |
| Forward Imputation | − | $0.4902 \pm 0.0000$ | $0.6148 \pm 0.0000$ |
| Single-Task GP | $0.8360 \pm 0.0013$ | $0.4167 \pm 0.0006$ | $0.3913 \pm 0.0002$ |
| Multi-Task GP | $0.8722 \pm 0.0015$ | $0.4121 \pm 0.0005$ | $0.3923 \pm 0.0008$ |
| HVAE RNN | $1.4380 \pm 0.0049$ | $0.7804 \pm 0.0073$ | $1.0382 \pm 0.0086$ |
| HVAE RNN-ODE | $1.3464 \pm 0.0036$ | $0.6864 \pm 0.0069$ | $0.8330 \pm 0.0093$ |
| HVAE ODE-RNN-ODE | $1.1533 \pm 0.0286$ | $0.5447 \pm 0.0228$ | $0.5642 \pm 0.0334$ |
| HTVAE mTAN | $1.0498 \pm 0.0013$ | $0.4931 \pm 0.0008$ | $0.4848 \pm 0.0008$ |
| **HeTVAE** | $\mathbf{0.6662 \pm 0.0023}$ | $\mathbf{0.3978 \pm 0.0003}$ | $\mathbf{0.3716 \pm 0.0001}$ |

a ODE-RNN encoder and neural ODE decoder. Finally, we compare to **HTVAE mTAN** (Shukla & Marlin, 2021a), a temporal VAE model consisting of multi-time attention networks producing homoscedastic output. For VAE models with homoscedastic output, we treat the output variance term as a hyperparameter and select the variance using log likelihood on the validation set. Architecture details for these methods can be found in Appendix A.4. As baselines, we also consider deterministic mean and forward imputation-based methods. **Forward imputation** always predicts the last observed value on each dimension, while **mean imputation** predicts the mean of all the observations for each dimension.

**Synthetic Data Results:** Figure 2 shows sample visualization output for the synthetic dataset. For this experiment, we compare HTVAE mTAN, the single task Gaussian process STGP, and the proposed HeTVAE model. We vary the number of observed points $(3, 10, 20)$ and each model is used to infer the distribution over the remaining time points. We draw multiple samples from the VAE latent state for HeTVAE and HTVAE mTAN and visualize the distribution of the resulting mixture. As we can see, the interpolations produced by HTVAE mTAN have approximately constant uncertainty across time and this uncertainty level does not change even when the number of points conditioned on increases. On the other hand, both HeTVAE and STGP show variable uncertainty across time. Their uncertainty reduces in the vicinity of input observations and increases in gaps between observations. Even though the STGP has an advantage in this experiment (the synthetic data were generated with an RBF kernel smoother and STGP uses RBF kernel as the covariance function), the proposed model HeTVAE shows comparable interpolation performance. We show more qualitative results in Appendix A.3.

**Real Data Results:** Tables 1, 2 and 3 compare the interpolation performance of all the approaches on PhysioNet, MIMIC-III and Climate dataset respectively. HeTVAE outperforms the prior approaches with respect to the negative log likelihood score on all three datasets. Gaussian Process based methods − STGP and MTGP achieve second and third best performance respectively. We emphasize that while the MAE and MSE values for some of the prior approaches are close to those obtained by the HeTVAE model, the primary metric of interest for comparing probabilistic interpolation approaches is log likelihood, where the HeTVAE performs much better than the other methods.

Table 3: Interpolation performance on Climate dataset.

| Model | Negative Log Likelihood | Mean Absolute Error | Mean Squared Error |
|---|---|---|---|
| Mean Imputation | − | $0.4539 \pm 0.0000$ | $0.8403 \pm 0.0000$ |
| Forward Imputation | − | $0.2979 \pm 0.0000$ | $0.8426 \pm 0.0000$ |
| Single-Task GP | $0.2478 \pm 0.0016$ | $\mathbf{0.2738 \pm 0.0002}$ | $\mathbf{0.4886 \pm 0.0001}$ |
| Multi-Task GP | − | − | − |
| HVAE RNN | $1.3666 \pm 0.0674$ | $0.4838 \pm 0.0474$ | $0.8587 \pm 0.0863$ |
| HVAE RNN-ODE | $1.1769 \pm 0.0032$ | $0.3514 \pm 0.0067$ | $0.6076 \pm 0.0059$ |
| HVAE ODE-RNN-ODE | $1.1766 \pm 0.0053$ | $0.3531 \pm 0.0034$ | $0.5953 \pm 0.0051$ |
| HTVAE mTAN | $0.9262 \pm 0.0073$ | $0.2916 \pm 0.0046$ | $0.5162 \pm 0.0060$ |
| **HeTVAE** | $\mathbf{0.1287 \pm 0.0242}$ | $0.2813 \pm 0.0034$ | $0.5013 \pm 0.0116$ |

Table 4: Ablation Study of HeTVAE: Negative Log Likelihood on PhysioNet and MIMIC-III.

| Model | PhysioNet | MIMIC-III |
|---|---|---|
| HeTVAE | $\mathbf{0.5542 \pm 0.0209}$ | $\mathbf{0.6662 \pm 0.0023}$ |
| HeTVAE - ALO | $0.6087 \pm 0.0136$ | $0.6869 \pm 0.0111$ |
| HeTVAE - DET | $0.6278 \pm 0.0017$ | $0.7478 \pm 0.0028$ |
| HeTVAE - INT | $0.6539 \pm 0.0107$ | $0.7430 \pm 0.0011$ |
| HeTVAE - HET - ALO | $1.1304 \pm 0.0016$ | $0.9272 \pm 0.0002$ |

We note that the MAE/MSE of the VAE-based models with homoscedastic output can be improved by using a small fixed variance during training. However, this produces even worse log likelihood values. Further, we note that the current implementation of MTGP is not scalable to the Climate dataset (270 dimensions). We provide experiments on an additional dataset in Appendix A.1.

**Ablation Results:** Table 4 shows the results of ablating several different components of the HeT-VAE model and training procedure. The first row shows the results for the full proposed approach. The HeTVAE - ALO ablation shows the result of removing the augmented learning objective and training the model only using the ELBO. This results in an immediate drop in performance on PhysioNet. HeTVAE - DET removes the deterministic pathway from the model, resulting in a performance drop on both MIMIC-III and PhysioNet. HeTVAE - INT removes the intensity encoding pathway from the UnTAND module. It results in a large drop in performance on both datasets. HeT-VAE - HET- ALO removes the heteroscedastic layer and the augmented learning objective (since the augmented learning objective is introduced to improve the learning in the presence of heteroscedastic layer), resulting in a highly significant drop on both datasets. These results show that all of the components included in the proposed model contribute to improved model performance. We provide more ablation results in Appendix A.2 and discuss hyperparameter selection in Appendix A.5.

## 5  DISCUSSION AND CONCLUSIONS

In this paper, we have proposed the Heteroscedastic Temporal Variational Autoencoder (HeTVAE) for probabilistic interpolation of irregularly sampled time series data. HeTVAE consists of an input sparsity-aware encoder, parallel deterministic and probabilistic pathways for propagating input uncertainty to the output, and a heteroscedastic output distribution to represent variable uncertainty in the output interpolations. Furthermore, we propose an augmented training objective to combat the presence of additional local optima that arise from the use of the heteroscedastic output structure. Our results show that the proposed model significantly improves uncertainty quantification in the output interpolations as evidenced by significantly improved log likelihood scores compared to several baselines and state-of-the-art methods. While the HeTVAE model can produce a probability distribution over an arbitrary collection of output time points, it is currently restricted to producing marginal distributions. As a result, sampling from the model does not necessarily produce smooth trajectories as would be the case with GPR-based models. Augmenting the HeTVAE model to account for residual correlations in the output layer is an interesting direction for future work.

## 6 REPRODUCIBILITY STATEMENT

The source code for reproducing the results in this paper is available at `https://github.com/reml-lab/hetvae`. It contains the instructions to reproduce the results in the paper including the hyperparameters. The hyperparameter ranges searched for each model are fully described in Appendix A.5. The source code also includes the synthetic dataset generation process as well as one of the real-world dataset. The other datasets can be downloaded and prepared following the preprocessing steps notes in Appendix A.6.1.

## ACKNOWLEDGEMENTS

Research reported in this paper was partially supported by the National Institutes of Health under award number 1P41EB028242.

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

## A    APPENDIX

### A.1    ADDITIONAL RESULTS

We also perform experiments on the UCI electricity dataset (described in Appendix A.6.1). We follow the same experiment protocols described in Section 4. As we can see from Table 5, the proposed model HeTVAE outperforms the prior approaches across all three metrics.

### A.2    ABLATION STUDY

Tables 6 and 7 show the complete results of ablating several different components of the HeTVAE model and training procedure with respect to all three evaluation metrics on PhysioNet and MIMIC-III respectively. We denote different components of the HeTVAE model as − HET: heteroscedastic output layer, ALO: augmented learning objective, INT: intensity encoding, DET: deterministic pathway. The results show selected individual and compound ablations of these components and indicate that all of these components contribute significantly to the model's performance in terms of the negative log likelihood score. We provide detailed comments below.

Table 5: Interpolation performance on Electricity Dataset.

| Model | Negative Log Likelihood | Mean Absolute Error | Mean Squared Error |
|---|---|---|---|
| Mean Imputation | – | $0.6765 \pm 0.0000$ | $1.0311 \pm 0.0000$ |
| Forward Imputation | – | $0.6163 \pm 0.0000$ | $1.2626 \pm 0.0000$ |
| Single-Task GP | $0.8972 \pm 0.0009$ | $0.5456 \pm 0.0007$ | $0.7827 \pm 0.0005$ |
| Multi-Task GP | $0.7767 \pm 0.0033$ | $0.5324 \pm 0.0016$ | $0.7782 \pm 0.0006$ |
| HVAE RNN | $1.3981 \pm 0.0043$ | $0.6267 \pm 0.0055$ | $0.9577 \pm 0.0093$ |
| HVAE RNN-ODE | $1.3947 \pm 0.0054$ | $0.6262 \pm 0.0074$ | $0.9469 \pm 0.0111$ |
| HVAE ODE-RNN-ODE | $1.4089 \pm 0.0095$ | $0.6453 \pm 0.0042$ | $0.9792 \pm 0.0184$ |
| HTVAE mTAN | $1.4040 \pm 0.0148$ | $0.6392 \pm 0.0250$ | $0.9724 \pm 0.0366$ |
| **HeTVAE** | $\mathbf{0.7055 \pm 0.0103}$ | $\mathbf{0.5049 \pm 0.0039}$ | $\mathbf{0.7503 \pm 0.0162}$ |

Table 6: Ablation Study of HeTVAE on PhysioNet.

| Model | Negative Log Likelihood | Mean Absolute Error | Mean Squared Error |
|---|---|---|---|
| HetVAE | $\mathbf{0.5542 \pm 0.0209}$ | $\mathbf{0.3911 \pm 0.0004}$ | $\mathbf{0.5778 \pm 0.0020}$ |
| HeTVAE - ALO | $0.6087 \pm 0.0136$ | $0.4087 \pm 0.0008$ | $0.6121 \pm 0.0063$ |
| HeTVAE - DET | $0.6278 \pm 0.0017$ | $0.4089 \pm 0.0005$ | $0.5950 \pm 0.0018$ |
| HeTVAE - INT | $0.6539 \pm 0.0107$ | $0.4013 \pm 0.0005$ | $0.5935 \pm 0.0015$ |
| HeTVAE - HET - ALO | $1.1304 \pm 0.0016$ | $0.3990 \pm 0.0003$ | $0.5871 \pm 0.0016$ |
| HeTVAE - DET - ALO | $0.7425 \pm 0.0066$ | $0.4747 \pm 0.0024$ | $0.6963 \pm 0.0031$ |
| HeTVAE - PROB - ALO | $0.7749 \pm 0.0047$ | $0.4251 \pm 0.0029$ | $0.6230 \pm 0.0040$ |
| HeTVAE - INT - DET - ALO | $0.7866 \pm 0.0029$ | $0.4857 \pm 0.0003$ | $0.7120 \pm 0.0007$ |
| HeTVAE - HET - INT - DET - ALO | $1.2426 \pm 0.0028$ | $0.5056 \pm 0.0004$ | $0.7167 \pm 0.0016$ |

**Effect of Heteroscedastic Layer:**  Since the augmented learning objective is introduced to improve the learning in the presence of heteroscedastic layer, we remove the augmented learning objective (ALO) with the heteroscedastic layer (HET). This ablation corresponds to HeTVAE - HET - ALO. As we can see from both Table 6 and 7, this results in a highly significant drop in the log likelihood performance as compared to the full HeTVAE model on both datasets. However, it results in only a slight drop in performance with respect to MAE and MSE, which is sensible as the HET component only affects uncertainty sensitive performance metrics.

**Effect of Intensity Encoding:**  HeTVAE - INT removes the intensity encoding pathway from the UnTAND module. It results in an immediate drop in performance on both datasets. We also compare the effect of intensity encoding after removing the deterministic pathway and the augmented learning objective. These ablations are shown in HeTVAE - DET - ALO and HeTVAE - INT - DET - ALO. The performance drop is less severe in this case because of the propensity of the heteroscedastic output layer to get stuck in poor local optima in the absence of the augmented learning objective (ALO).

**Effect of Augmented Learning Objective:**  The HeTVAE - ALO ablation shows the result of removing the augmented learning objective and training the model only using only the ELBO. This results in an immediate drop in performance on PhysioNet. The performance drop is less severe on MIMIC-III. We further perform this ablation without the DET component and observe severe drops in performance across all metrics on both datasets. These ablations correspond to HeTVAE - DET and HeTVAE - DET - ALO. This shows that along with ALO component, the DET component also constrains the model from getting stuck in local optima where all of the structure in the data is explained as noise. We show interpolations corresponding to these ablations in Appendix A.3.1.

**Effect of Deterministic Pathway:**  HeTVAE - DET removes the deterministic pathway from the model, resulting in a performance drop on both MIMIC-III and PhysioNet across all metrics. We further compare the performance of both the probabilistic and deterministic pathways in isolation as shown by ablation HeTVAE - DET - ALO and HeTVAE - PROB - ALO. We observe that the

Table 7: Ablation Study of HeTVAE on MIMIC-III.

| Model | Negative Log Likelihood | Mean Absolute Error | Mean Squared Error |
|---|---|---|---|
| HetVAE | **0.6662 ± 0.0023** | **0.3978 ± 0.0003** | **0.3716 ± 0.0001** |
| HeTVAE - ALO | 0.6869 ± 0.0111 | 0.4043 ± 0.0006 | 0.3840 ± 0.0007 |
| HeTVAE - DET | 0.7478 ± 0.0028 | 0.4129 ± 0.0008 | 0.3845 ± 0.0009 |
| HeTVAE - INT | 0.7430 ± 0.0011 | 0.4066 ± 0.0001 | 0.3837 ± 0.0001 |
| HeTVAE - HET - ALO | 0.9272 ± 0.0002 | 0.4044 ± 0.0001 | 0.3765 ± 0.0001 |
| HeTVAE - DET - ALO | 0.9005 ± 0.0052 | 0.5177 ± 0.0004 | 0.5325 ± 0.0008 |
| HeTVAE - PROB - ALO | 0.7472 ± 0.0056 | 0.4049 ± 0.0006 | 0.3833 ± 0.0008 |
| HeTVAE - INT - DET - ALO | 0.9245 ± 0.0021 | 0.5208 ± 0.0009 | 0.5358 ± 0.0012 |
| HeTVAE - HET - INT - DET - ALO | 1.0498 ± 0.0013 | 0.4931 ± 0.0008 | 0.4848 ± 0.0008 |

deterministic pathway HeTVAE - PROB - ALO outperforms the probabilistic pathway HeTVAE - DET - ALO in terms of log likelihood on MIMIC-III while the opposite is true in case of PhysioNet. However, on both datasets using only the deterministic pathway (HeTVAE - PROB - ALO) achieves better MAE and MSE scores as compared to using only the probabilistic pathway (HeTVAE - DET - ALO).

## A.3 VISUALIZATIONS

### A.3.1 INTERPOLATIONS ON PHYSIONET

Figure 3 shows example interpolations on the PhysioNet dataset. Following the experimental setting mentioned in Section 4, the models were trained using all dimensions and the inference uses all dimensions. We only show interpolations corresponding to Heart Rate as an illustration. As we can see, the STGP and HeTVAE models exhibit good fit and variable uncertainty on the edges where there are no observations. We can also see that mTAN trained with homoscedastic output is not able to produce as good a fit because of the fixed variance at the output (discussed in Section 4).

The most interesting observation is the performance of HeTVAE - DET - ALO, an ablation of HeTVAE model that retains heteroscedastic output, but removes the deterministic pathways and the augmented learning objective. This ablation significantly underfits the data and performs similar to mTAN. This is an example of local optima that arises from the use of a heteroscedastic output layer where the mean is excessively smooth and all of the structure in the data is explained as noise. We address this with the use of augmented learning objective described in Section 3.3. As seen in the Figure 3, adding the augmented learning objective (HeTVAE - DET) clearly improves performance.

### A.3.2 SYNTHETIC DATA VISUALIZATIONS: SPARSITY

In this section, we show supplemental interpolation results on the synthetic dataset. The setting here is same as in Section 4. Figure 4 compares HTVAE mTAN, the single task Gaussian process STGP, the proposed HeTVAE model and an ablation of proposed model without intensity encoding HeTVAE - INT. We vary the number of observed points $(3, 10, 20)$ and each model is used to infer the distribution over the remaining time points. We draw multiple samples from the VAE latent state for HeTVAE, HeTVAE - INT and HTVAE mTAN, and visualize the distribution of the resulting mixture. Figure 4 illustrates the interpolation performance of each of the models. As we can see, the interpolations produced by HTVAE mTAN have approximately constant uncertainty across time and this uncertainty level does not change even when the number of points conditioned on increases. On the other hand, both HeTVAE and STGP show variable uncertainty across time. Their uncertainty reduces in the vicinity of input observations and increases in gaps between observations. The HeTVAE-INT model performs slightly better than HTVAE mTAN model but it does not show variable uncertainty due to input sparsity like HeTVAE.

### A.3.3 SYNTHETIC DATA VISUALIZATIONS: INTER-OBSERVATION GAP

To demonstrate the effectiveness of intensity encoder (INT), we perform another experiment on synthetic dataset where we increase the maximum inter-observation gap between the observations.

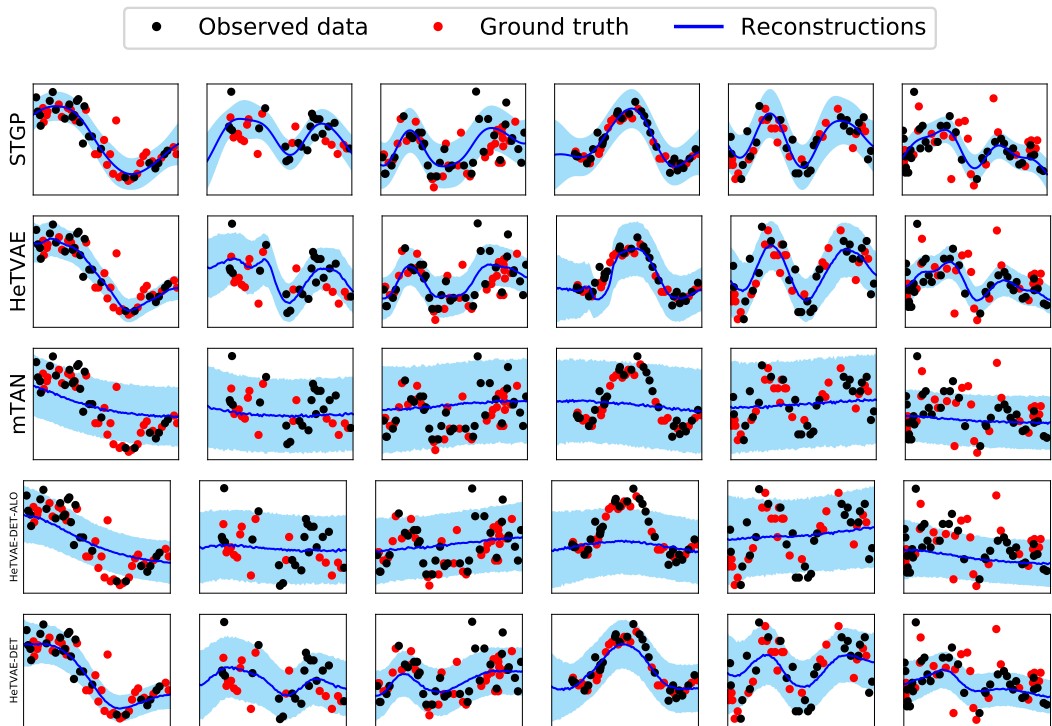

Figure 3: In this figure, we show example interpolations of one dimension corresponding to Heart Rate on the PhysioNet dataset. The columns correspond to different examples. The rows correspond to STGP, HeTVAE, HTVAE mTAN, HeTVAE-DET-ALO and HeTVAE-DET respectively. The shaded region corresponds to $\pm$ one standard deviation. STGP, HeTVAE and HeTVAE-DET exhibit variable output uncertainty and good fit while mTAN and HETVAE-DET-ALO does not.

We follow the same training protocol as described in Section 4. At test time, we condition on 10 observed points with increasing maximum inter-observation gap. We vary the maximum inter-observation gap from $20\%$ to $80\%$ of the length of the original time series. Each model is used to infer single time point marginal distributions over values at the rest of the available time points in the test instance.

Figure 5 shows the interpolations with increasing maximum inter-observation gap. STGP and HeTVAE show variable uncertainty with time and the uncertainty increases with increasing maximum inter-observation gap. On the other hand, HTVAE mTAN with homoscedastic output shows approximately constant uncertainty with time and also across different maximum inter-observation gaps. These results clearly show that HTVAE mTAN produces over-confident probabilistic interpolations over large gaps.

Furthermore, we show an ablation of the proposed model HeTVAE - INT, where we remove the intensity encoder and perform the interpolations. As we see from the figure, this leads to approximately constant uncertainty across time as well as different maximum inter-observation gaps. This shows that the HeTVAE model is not able to capture uncertainty due to input sparsity as effectively without the intensity encoder.

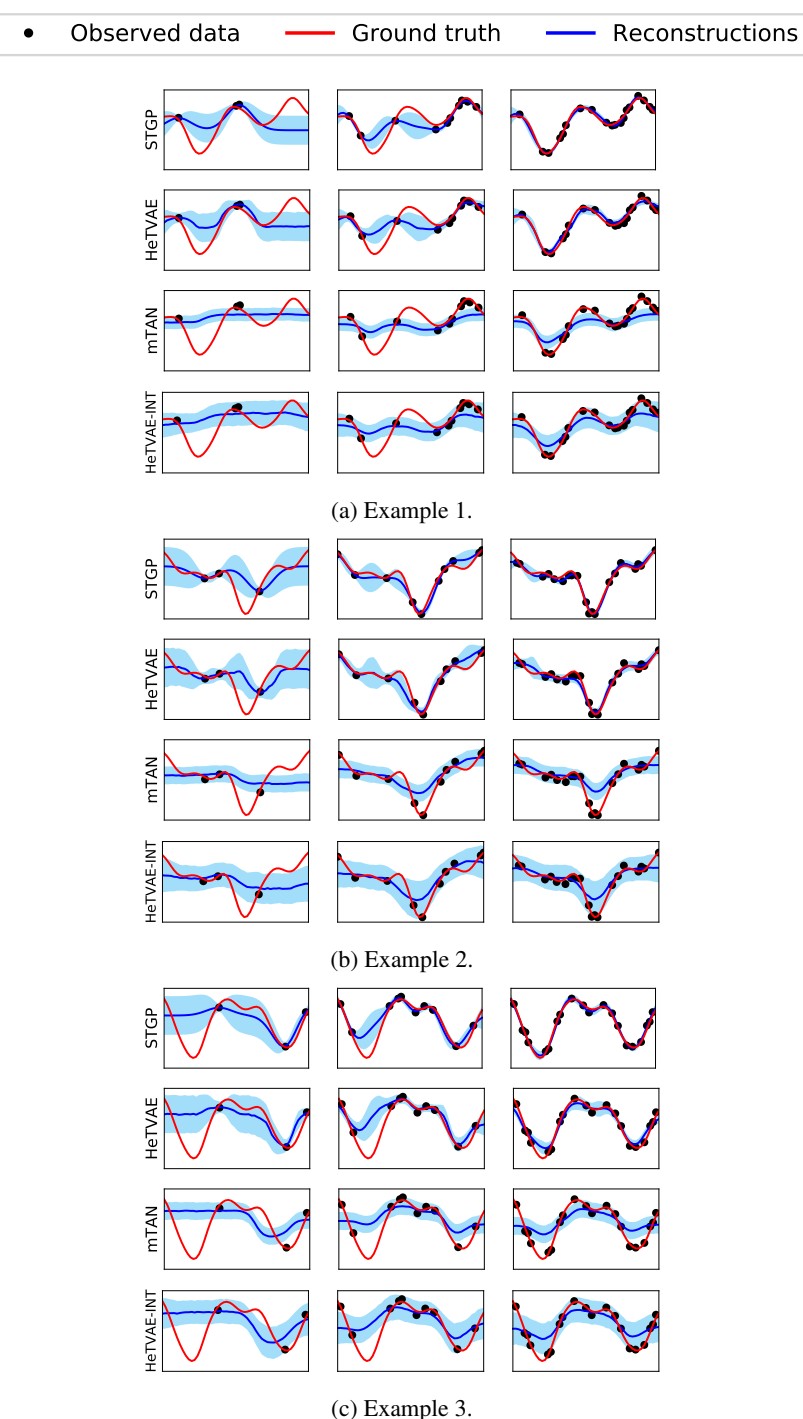

Figure 4: Additional interpolation results on the synthetic dataset. The 3 columns correspond to interpolation results with increasing numbers of observed points: 3, 10 and 20 respectively. The shaded region corresponds to ± one standard deviation. STGP and HeTVAE exhibit variable output uncertainty in response to input sparsity while mTAN and HeTVAE - INT do not.

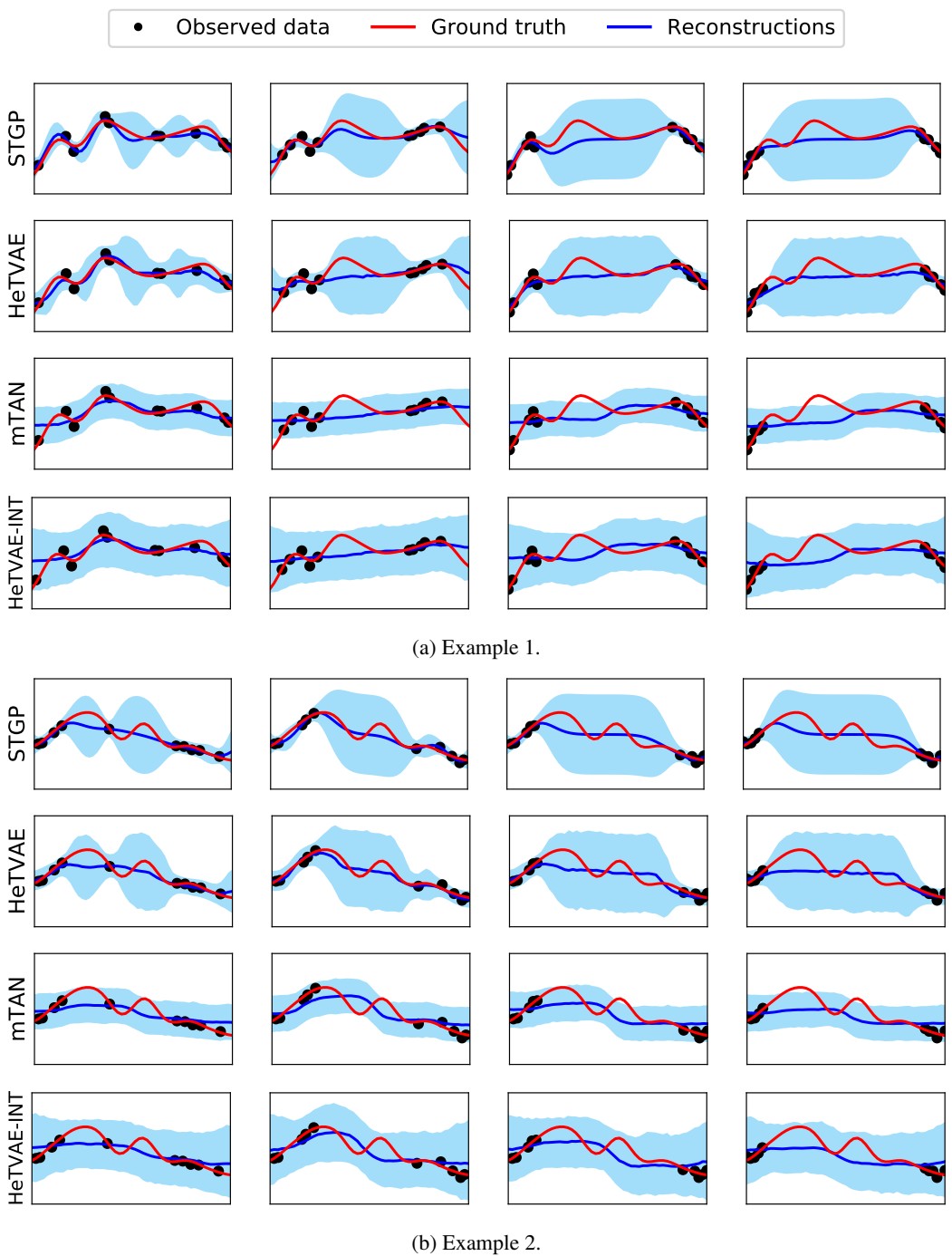

Figure 5: In this figure, we show example interpolations on the synthetic dataset with increasing maximum inter-observation gap. The columns correspond to an inter-observation gap of size 20%, 40%, 60% and 80% of the length of original time series. The rows correspond to STGP, HeT-VAE, HTVAE mTAN and HeTVAE-INT respectively. The shaded region corresponds to the confidence region. STGP and HeTVAE exhibit variable output uncertainty while mTAN and HeTVAE-INT does not.

## A.4 Architecture Details

**HeTVAE:** Learnable parameters in the UnTAND architecture shown in Figure 1a include the weights of the three linear layers and the parameters of the shared time embedding functions. Each time embedding function is a one layer fully connected network with a sine function non-linearity. The two linear layers on top of embedding function are linear projections from time embedding dimension $d_e$ to $d_e/H$ where $H$ is the number of time embeddings. Note that these linear layers do not share parameters. The third linear layer performs a linear projection from $2 * D * H$ to $J$. It takes as input the concatenation of the VAL encoder output and INT encoder output and produces an output of dimension $J$. $d_e$, $H$ and $J$ are all hyperparameters of the architecture. The ranges considered are described in the next section.

The HeTVAE model shown in the Figure 1b consists of three MLP blocks apart from the UnTAND modules. The MLP in the deterministic path is a one layer fully connected layer that projects the UnTAND output to match the dimension of the latent state. The remaining MLP blocks are two-layer fully connected networks with matching width and ReLU activations. The MLP in the decoder takes the output of UnTAND module and outputs the mean and variance of dimension $D$ and sequence length $\mathbf{t}'$. We use a softplus transformation on the decoder output to get the variance $\boldsymbol{\sigma}_i = 0.01 + \texttt{softplus}(f_\sigma^{dec}(\mathbf{h}_i^{dec}))$. Similarly, in the probabilistic path, we apply an exponential transformation to get the variance of the $q$ distribution $\boldsymbol{\sigma}_k^2 = \exp(f_\sigma^{enc}(\mathbf{h}_k^{enc}))$. We use $K$ reference time points regularly spaced between $0$ and $1$. $K$ is considered to be a hyperparameter of the architecture. The ranges considered are described in the next section.

**Baselines:** For the HTVAE mTAN, we use a similar architecture as HeTVAE where we remove the deterministic path, heteroscedastic output layer and use the mTAND module instead of the UnTAND module (Shukla & Marlin, 2021a). We use the same architectures for the ODE and RNN-based VAEs as Rubanova et al. (2019).

## A.5 Hyperparameters

**HeTVAE:** We fix the time embedding dimension to $d_e = 128$. The number of embeddings $H$ is searched over the range $\{1, 2, 4\}$. We search the number of reference points $K$ over the range $\{4, 8, 16, 32\}$, the latent dimension over the range $\{8, 16, 32, 64, 128\}$, the output dimension of Un-TAND $J$ over the range $\{16, 32, 64, 128\}$, and the width of the two-layer fully connected layers over $\{128, 256, 512\}$. In augmented learning objective, we search for $\lambda$ over the range $\{1.0, 5.0, 10.0\}$. We use the Adam Optimizer for training the models. Experiments are run for $2,000$ iterations with a learning rate of $0.0001$ and a batch size of $128$. The best hyperparameters are reported in the code. We use $100$ samples from the probabilistic latent state to compute the evaluation metrics.

**Ablations:** We note that the ablations were not performed with a fixed architecture. For all the ablation models, we tuned the hyperparameters and reported the results with the best hyperparameter setting. We also made sure that the hyperparameter ranges for ablated models with just deterministic/probabilistic path were wide enough that the optimal ablated models did not saturate the end of the ranges for architectural hyper-parameter values including the dimensionality of the latent representations.

**VAE Baselines:** For VAE models with homoscedastic output, we treat the output variance term as a hyperparameter and select the variance over the range $\{0.01, 0.1, 0.2, 0.4, 0.6, 0.8, 1.0, 1.2, 1.4, 1.6, 1.8, 2.0\}$. For HTVAE mTAN, we search the corresponding hyperparameters over the same range as HeTVAE. For ODE and RNN based VAEs, we search for GRU hidden units, latent dimension, the number of hidden units in the fully connected network for the ODE function in the encoder and decoder over the range $\{20, 32, 64, 128, 256\}$. For ODEs, we also search the number of layers in fully connected network in the range $\{1, 2, 3\}$. We use a batch size of $50$ and a learning rate of $0.001$. We use $100$ samples from the latent state to compute the evaluation metrics.

**Gaussian Processes:** For single task GP, we use a squared exponential kernel. In case of multi-task GP, we experimented with the Matern kernel with different smoothness parameters, and the

squared exponential kernel. We found that Matern kernel performs better. We use maximum marginal likelihood to train the GP hyperparameters. We search for learning rate over the range $\{0.1, 0.01, 0.001\}$ and run for $100$ iterations. We search for smoothness parameter over the range $\{0.5, 1.5, 2.5\}$. We search for the batch size over the range $\{32, 64, 128, 256\}$.

## A.6  TRAINING DETAILS

### A.6.1  DATA GENERATION AND PREPROCESSING

**Synthetic Data Generation:**    We generate a synthetic dataset consisting of $2000$ trajectories each consisting of $50$ time points with values between $0$ and $1$. We fix $10$ reference time points and draw values for each from a standard normal distribution. We then use an RBF kernel smoother with a fixed bandwidth of $\alpha = 120.0$ to construct local interpolations over the $50$ time points. The data generating process is shown below:

$$z_k \sim \mathcal{N}(0,1), k \in [1, \cdots, 10]$$
$$r_k = 0.1 * k$$
$$t_i = 0.02 * i, i \in [1, \cdots, 50]$$
$$x_i = \frac{\sum_k \exp(-\alpha(t_i - r_k)^2) \cdot z_k}{\sum_{k'} \exp(-\alpha(t_i - r_{k'})^2)} + \mathcal{N}(0, 0.1^2)$$

We randomly sample $3 - 10$ observations from each trajectory to simulate a sparse and irregularly sampled univariate time series.

**PhysioNet:**    The PhysioNet Challenge 2012 dataset (Silva et al., 2012) consists of multivariate time series data with $37$ physiological variables from intensive care unit (ICU) records. Each record contains measurements from the first $48$ hours after admission. We use the protocols described in Rubanova et al. (2019) and round the observation times to the nearest minute resulting in $2880$ possible measurement times per time series. The data set consists includes $8000$ instances that can be used for interpolation experiments. PhysioNet is freely available for research use and can be downloaded from `https://physionet.org/content/challenge-2012/`.

**MIMIC-III:**    The MIMIC-III data set (Johnson et al., 2016) is a multivariate time series dataset containing sparse and irregularly sampled physiological signals collected at Beth Israel Deaconess Medical Center. We use the procedures proposed by Shukla & Marlin (2019) to process the data set. This results in $53,211$ records each containing $12$ physiological variables. We use all $53,211$ instances to perform interpolation experiments. MIMIC-III is available through a permissive data use agreement which can be requested at `https://mimic.mit.edu/iii/gettingstarted/`. Once the request is approved, the dataset can be downloaded from `https://mimic.mit.edu/iii/gettingstarted/dbsetup/`. The instructions and code to extract the MIMIC-III dataset is given at `https://github.com/mlds-lab/interp-net`.

**Climate Dataset:**    The U.S. Historical Climatology Network Monthly (USHCN) dataset (Menne et al., 2016) is a publicly available dataset consisting of daily measurements of $5$ climate variables $-$ daily maximum temperature, daily minimum temperature, whether it was a snowy day or not, total daily precipitation, and daily snow precipitation. It contains data from the last $150$ years for $1,218$ meteorological stations scattered over the United States. Following the preprocessing steps of Che et al. (2018b), we extract daily climate data for $100$ consecutive years starting from $1910$ to $2009$ from $54$ stations in California. To get multi-rate time series data, we split the stations into $3$ groups with sampling rates of $2$ days, $1$ week, and $1$ month respectively. We divide the data into smaller time series consisting of yearly data and end up with a dataset of $100$ examples each consisting of $270$ features. We perform the interpolation task on this dataset where we compute the feature values every day using the multi-rate time series data. The dataset is available for download at `https://cdiac.ess-dive.lbl.gov/ftp/ushcn_daily/`.

**Electricity Dataset:**    The UCI household electricity dataset contains measurements of seven different quantities related to electricity consumption in a household. The data are recorded every minute for $47$ months between December 2006 and November 2010, yielding over 2 million observations. To simulate irregular sampling, we keep observations only at durations sampled from

an exponential distribution with $\lambda = 20$. Following the preprocessing step of Binkowski et al. (2018), we also do random feature sampling where we choose one out of seven features at each time step. We divide the data into smaller time series consisting of monthly data and end up with a dataset of 1431 examples each consisting of 7 features. We perform interpolation experiments on this dataset where we compute feature values every minute using the irregularly sampled data. The dataset is available for download at `https://archive.ics.uci.edu/ml/datasets/individual+household+electric+power+consumption`.

**Dataset Preprocessing:**   We rescale time to be in $[0, 1]$ for all datasets. We also re-scale all dimensions. In case of PhysioNet and MIMIC-III, for each dimensions we first remove outliers in the outer $0.1\%$ percentile region. We then compute the mean and standard deviation of all observations on that dimension. The outlier detection step is used to mitigate the effect of rare large values in the data set from affecting the normalization statistics. Finally, we z-transform all of the available data (including the points identified as outliers). No data points are discarded from the data sets during the normalization process.

### A.6.2   SOURCE CODE

The source code for reproducing the results in this paper is available at `https://github.com/reml-lab/hetvae`.

### A.6.3   COMPUTING INFRASTRUCTURE

All experiments were run on a Nvidia Titan X and 1080 Ti GPUs. The time required to run all the experiments in this paper including hyperparameter tuning was approximately eight days using eight GPUs.

