# OpenReview forum: "Heteroscedastic Temporal Variational Autoencoder For Irregularly Sampled Time Series"
_ICLR.cc/2022/Conference — ICLR 2022 Poster_

### Official Review · Reviewer_7Ard · 2021-10-25

**Correctness:** 4
**Technical Novelty And Significance:** 3
**Empirical Novelty And Significance:** 3
**Recommendation:** 5
**Confidence:** 4

**Main Review:**

Clarity: I am confused about the differences between heteroscedastic and homoscedastic after reading the paper. Does the heteroscedastic output means that you learn the variance and the homoscedastic output means you fix the variance? What is the shape of w and v in equation 3? what is g in equation 7?

Originality: I think the intensity encoding is novel and makes sense to make the model be aware of input sparsity. For the heteroscedastic output layer, based on my understanding (see Clarity above), it just allows the model to learn the variance rather than using a fixed variance. If my understanding is correct, I don't this heteroscedastic output layer is novel because there are lots of works that learn a variance for maximum likelihood estimation. Combining a deterministic path and a probabilistic path is not novel, which has been proposed in [1, 2]. The augmented training objective in equation 11 that encourages the predicted mean to be close to samples x makes sense, but I think there should be some existing works that also use this trick.

Experiments: I am satisfied with the performance this model achieves. However, I think the authors should also compare to Neural Process [1] and Attentive Neural Process [2], which are mentioned in the related work. [1,2] also use attention mechanism and do probabilistic interpolation. This paper also misses citation and comparison to a previous work NRTSI [3] that can also impute irregularly-sampled time series. I suggest the author compare to NRTSI on the irregularly-sampled Billiard dataset introduced in NRTSI.  Also, I am not clear about what the baseline HTVAE mTAN means? Does it mean "homoscedastic temporal VAE mTAN"? If my understanding is correct, I think the author should also compare to HeTVAE mTAN ("heteroscedastic temporal VAE mTAN") that allows mTAN to learn a variance rather than using a fixed variance. For the ablation study model HeTVAE - DET, because the deterministic path is removed, the model capacity decreases and makes the comparison unfair. For a fair comparison, I suggest increasing the capacity of the probabilistic path after removing the deterministic path.



[1] Garnelo, Marta, et al. "Neural processes." arXiv preprint arXiv:1807.01622 (2018).

[2] Kim, Hyunjik, et al. "Attentive neural processes." arXiv preprint arXiv:1901.05761 (2019).

[3] Shan, Siyuan, and Junier B. Oliva. "NRTSI: Non-Recurrent Time Series Imputation." arXiv preprint arXiv:2102.03340 (2021).



**Summary Of The Paper:**

This paper introduces several improvements over the previous work mTAN to better support probabilistic interpolation. Specifically, intensity encoding is introduced to make the model be aware of information about input sparsity. Also, the homoscedastic output distribution used by previous work is replaced by a heteroscedastic distribution. Experiments results show that this improved model (HeTVAE) achieves both better likelihood estimation and mean prediction compared to previous works.

**Summary Of The Review:**

Given the limited novelty, some unclarities, and lacks of some baselines, I give 5 to this paper now. If the authors can clarify and add these baseline results, I may consider increasing my score.

---

> ### Author Response · Authors · 2021-11-15
> **Response to Reviewer 7Ard (Part 1)**
>
> Thank you for the detailed review. Below we address comments.
>
> *Q: I am confused about the differences between heteroscedastic and homoscedastic after reading the paper. Does the heteroscedastic output mean that you learn the variance and the homoscedastic output means you fix the variance?*
>
> A: Yes, the homoscedastic output means a constant variance per time point for each example. The heteroscedastic output layer means that the model outputs variable variance per time point for each example. We achieve this by employing a separate head (corresponding to the variance)  in addition to the mean output as shown in Figure 1b.
>
> *Q: What is the shape of w and v in equation 3? what is g in equation 7?*
>
> A: The parameters w and v are each d_e × d_e/H matrices where H is the number of time embeddings. We have described this in architecture details (Appendix A.4). g is a feed-forward network that takes in as input the UnTAND output embeddings h_enc and produces a deterministic temporal representation (at each reference point) of the same dimension as the probabilistic latent state. We have defined g in the first paragraph of Section 3.3.
>
> *Q: I think the intensity encoding is novel and makes sense to make the model be aware of input sparsity. For the heteroscedastic output layer, based on my understanding (see Clarity above), it just allows the model to learn the variance rather than using a fixed variance. If my understanding is correct, I don't this heteroscedastic output layer is novel because there are lots of works that learn a variance for maximum likelihood estimation. Combining a deterministic path and a probabilistic path is not novel, which has been proposed in [1, 2]. The augmented training objective in equation 11 that encourages the predicted mean to be close to samples x makes sense, but I think there should be some existing works that also use this trick.*
>
> A: We thank the reviewer for acknowledging the novelty of the intensity encoding. We address the novelty concern with the heteroscedastic output layer below:
> - Firstly, there are limited works [1,2] that consider learning the variance for the maximum likelihood estimation but these approaches are not applicable for time series or irregularly sampled time series. To the best of our knowledge, this is the first paper that learns output variance for time series data and interprets the variance in a meaningful way.
> - Secondly, we show that just using the heteroscedastic output layer is not enough to learn meaningful and interpretable output uncertainties.  In fact, we show that the use of heteroscedastic output representation can lead the model to get stuck in poor local optima where the mean is essentially flat and all of the structure in the data is explained as noise (discussed in Appendix A.3.1).
> We propose an augmented learning objective to address the underfitting of predictive variance caused by the use of the heteroscedastic layer, a problem also noted in [1] and [2]. We verify this claim through several experiments and sample interpolations showing that the use of this augmented training objective has a strong positive impact on final model performance. To the best of our knowledge, this is the first work to use an augmented learning objective to learn variance in the heteroscedastic layer. We would like to ask the reviewer to point us to any specific references using this technique.

---

> > ### Author Response · Authors · 2021-11-15
> > **Response to Reviewer 7Ard (Part 2)**
> >
> > *Q:  I am not clear about what the baseline HTVAE mTAN means? Does it mean "homoscedastic temporal VAE mTAN"? If my understanding is correct, I think the author should also compare to HeTVAE mTAN ("heteroscedastic temporal VAE mTAN") that allows mTAN to learn a variance rather than using a fixed variance.*
> >
> > A: Yes, HTVAE mTAN refers to homoscedastic temporal VAE mTAN. We have already included the results with HeTVAE mTAN (heteroscedastic temporal VAE mTAN) in the paper. This model refers to HeTVAE - INT - DET - ALO in Tables 6 and 7 in the ablation study (Appendix A.2). We include the results below.
> >
> > Results on PhysioNet dataset:
> >
> > | **Model**     |  **Negative Log-Likelihood** | Mean Absolute Error |	Mean Squared Error |
> > | ----------- | ----------- | ----------- | ----------- |
> > | HeTVAE	 | ${\bf 0.5542	\pm	0.0209}$	|	${\bf	0.3911	\pm	0.0004	}$	|	${\bf	0.5778	\pm	0.002	}$ |
> > | HeTVAE - INT - DET - ALO(HeTVAE mTAN) 	 |	$	0.7866	\pm	0.0029	$	|	$	0.4857	\pm	0.0003	$	|	$	0.7120	\pm	0.0007	$ |
> >
> > Results on MIMIC-III dataset:
> >
> > | **Model**     |  **Negative Log-Likelihood** | Mean Absolute Error |	Mean Squared Error |
> > | ----------- | ----------- | ----------- | ----------- |
> > | HeTVAE	 | ${\bf	0.6662	\pm	0.0023	}$	|	${\bf	0.3978	\pm	0.0003	}$	|	${\bf	0.3716	\pm	0.0001	}$|
> > | HeTVAE - INT - DET - ALO(HeTVAE mTAN) 	 |	$	0.9245	\pm	0.0021	$	|	$	0.5208	\pm	0.0009	$	|	$	0.5358	\pm	0.0012	$|
> >
> > We can see that the proposed model performs significantly better than the HeTVAE mTAN across all three metrics on both methods. We provide more ablation in Tables 6 and 7 in Appendix A.2 showing that all of the components in the proposed model contribute significantly to the model’s performance in terms of the negative log-likelihood score.
> >
> > *Q: For the ablation study model HeTVAE - DET, because the deterministic path is removed, the model capacity decreases and makes the comparison unfair. For a fair comparison, I suggest increasing the capacity of the probabilistic path after removing the deterministic path.*
> >
> > A: We actually perform a fair comparison by increasing capacity in cases where we remove a part of the architecture as noted in Appendix A.5. We include the excerpt from the Appendix here: "We note that the ablations were not performed with a fixed architecture. For all the ablation models, we tuned the hyperparameters and reported the results with the best hyperparameter setting. We also made sure that the hyperparameter ranges for ablated models with just deterministic/probabilistic paths were wide enough that the optimal ablated models did not saturate the end of the ranges for architectural hyper-parameter values including the dimensionality of the latent representations."
> >
> > *Q: I am satisfied with the performance this model achieves. However, I think the authors should also compare to Neural Process [1] and Attentive Neural Process [2], which are mentioned in the related work. [1,2] also use attention mechanism and do probabilistic interpolation. This paper also misses citation and comparison to a previous work NRTSI [3] that can also impute irregularly-sampled time series.*
> >
> > A: As we point out in the related work the neural process approaches [1,2] are not applicable for irregularly sampled time series. It is also not clear how these approaches can be applied to sequential data. We would like to thank the reviewer for pointing out the related work [3]. We will add a discussion in the related work.
> >
> >
> > References:
> > 1. Garnelo, Marta, et al. "Neural processes." arXiv preprint arXiv:1807.01622 (2018).
> > 2. Kim, Hyunjik, et al. "Attentive neural processes." arXiv preprint arXiv:1901.05761 (2019).
> > 3. Shan, Siyuan, and Junier B. Oliva. "NRTSI: Non-Recurrent Time Series Imputation." arXiv preprint arXiv:2102.03340 (2021).

---

> > ### Comment · Reviewer_7Ard · 2021-11-19
> > **A exisiting work called NRTSI has already learnt output variance for time series data.**
> >
> > It seems that your work is not the first to learn output variance for time series data. An existing work NRTSI [3] already did this (see equation 5 of the NRTSI paper and the stochastic imputation results in the Figure 5 of NRTSI). NRTSI can perform stochastic imputation by learning the variance without any fancy tricks such as the "augmented learning objective" proposed in your paper.
> >
> > [3] Shan, Siyuan, and Junier B. Oliva. "NRTSI: Non-Recurrent Time Series Imputation." arXiv preprint arXiv:2102.03340 (2021).

---

> > > ### Author Response · Authors · 2021-11-19
> > > **HeTVAE focuses on probabilistic interpolation while NRTSI focuses on the deterministic interpolation of time series.**
> > >
> > > Firstly, we want to clarify that the comment "To the best of our knowledge, this is the first work to use an augmented learning objective to learn VAEs in the presence of the heteroscedastic layer " was made in response to the reviewer's comment "The augmented training objective in equation 11 that encourages the predicted mean to be close to samples x makes sense, but I think there should be some existing works that also use this trick". And this contribution mainly addresses the learning of VAE in the presence of a heteroscedastic output layer. As we have shown through several ablations (Appendix A.2 and A.3.1) and through several prior works [1,2] that naively attempting to learn the variance in a Gaussian decoder (in VAE) suffers from underfitting of predictive variance and can lead to poor results [3, 4, 5].
> > >
> > > Secondly, the NRTSI is not a VAE based approach, its more like a autoregressive prediction style model (without the recurrent structure). Another difference is that the NRTSI approach does not evaluate the predicted variance and they focus mainly on the deterministic evaluation metrics such as MAE/MSE which are not indicative of the uncertainty quantification. Also, looking at some of the stochastic trajectories generated by the model (in Fig 10 of [3]), it seems that variance has totally collapsed and the it is not as expressive as the probabilistic interpolations generated by HeTVAE. To be fair though, the NRTSI approach focuses on deterministic interpolation as can be seen by their evaluation metrics. On the other hand, our approach HeTVAE focuses on the probabilistic interpolation problem.
> > >
> > > Finally, we would like to thank the author for pointing out this relevant paper. We will add the NRTSI work in our related work and will also include this discussion there.
> > >
> > > References:
> > >
> > > 1. Garnelo, Marta, et al. "Neural processes." arXiv preprint arXiv:1807.01622 (2018).
> > > 2. Kim, Hyunjik, et al. "Attentive neural processes." arXiv preprint arXiv:1901.05761 (2019).
> > > 3. Rybkin, Oleh, Kostas Daniilidis, and Sergey Levine. "Simple and effective VAE training with calibrated decoders." ICML 2021.
> > > 4. Rezende, D. J. and Viola, F. Taming vaes. arXiv preprint arXiv:1810.00597, 2018.
> > > 5. Dai, B. and Wipf, D. Diagnosing and enhancing vae models. arXiv preprint arXiv:1903.05789, 2019.

---

### Official Review · Reviewer_EXpk · 2021-10-27

**Correctness:** 4
**Technical Novelty And Significance:** 3
**Empirical Novelty And Significance:** 3
**Recommendation:** 8
**Confidence:** 4

**Main Review:**

STRENGTHS

1. A great deal of modern machine learning literature for time series data often assumes regularly sampled time series, with no missing data and fixed size outputs. This is however rarely the case in many real world applications, which introduces many technical challenges that are not obvious to overcome with more standard architectures. The model presented in this paper is a possible way to tackle these challenges, and as such, I found this paper a very interesting read.
1. Despite building heavily on the mTAN model, the new ideas introduced in the paper are novel and well motivated.
2. Empirically, the HeTVAE outperforms competing methods by a large margin, and seem to be able to correctly capture uncertainty over time (as seen for example in figures 2 and 3)
3. The paper has extensive ablation studies that justify all the new components of the model

WEAKNESSES
1. I found the theoretical exposition in section 3 somewhat confusing in its current form, I could only follow it after reading details in the appendix and reading section 3 once again.
    1. Reference points play a key role in the HeTVAE, but from section 3 it is not clear what their role is as well as how they are chosen. I was only able to really grasp their role after reading the appendix and looking at the code, which should not be the case. Only in appendix A4 I could understand that they are regularly spaced in [0,1], and only in A.6.1 that the time is scaled between 0 and 1 in all datasets (after which the choice of reference points makes sense)
    2. related to the above, "reference points" are mentioned in the "intensity encoding" and "model output" paragraphs. But for a reader not familiar with the mTAN it is not obvious why we are interested in them.
    3. The prior over z is not defined in the paper
2. The "augmented learning objective" seems quite hacky to me, and I wonder if there are better ways to achieve the same (e.g. better initializations, KL annealing, ..).
    1. Learning output variances is normally not a problem in VAEs, why is it a problem in this case?
    2. How robust is the choice of the scaling factor lambda across different datasets?
3. Choosing equally spaced reference points means that if one uses as a test point a sample temporally close to the input data (or even a sample from the input data), the imputation of the model might be unnecessarily poor. Could this be improved?
4. As stated by the authors, the model is only able to provide a marginal distribution at each time step, which practically means that the sampled trajectories might look inconsistent (non-smooth). Exploring the usage of sequential latent variable models for this would be an interesting future research direction.
5. The baselines for HVAE RNN  and HVAE RNN-ODE are much worse than forward imputation, which makes me question the implementation of the models.

**Summary Of The Paper:**

This paper introduces a VAE-based model for interpolation of irregularly sampled time series. The temporal input data is mapped to a latent representation over fixed reference points with an attention mechanism, using an intensity network that allows to encode data sparsity information. This latent representation can then be used to interpolate points at new time steps. Thanks to the intensity network and the heteroscedastic output layer, the proposed HeTVAE model can capture uncertainty estimates over the interpolated points.
The model is tested on a number of datasets containing irregularly sampled points, and outperforms competing methods in the interpolation task.

**Summary Of The Review:**

This is a paper that can have an impact in real world applications, and as such I think it should be accepted. However, I did not find the paper easy to follow in this form.
For this, for now I have set "marginally above the acceptance threshold", but I would be happy to revise the score upwards if I see that the authors improve the exposition in the paper.

---

> ### Author Response · Authors · 2021-11-16
> **Response to Reviewer EXpk**
>
> Thank you for the thoughtful and detailed review. We address the concerns below:
>
> *Q: Reference points play a key role in the HeTVAE, but from section 3 it is not clear what their role is as well as how they are chosen. I was only able to really grasp their role after reading the appendix and looking at the code, which should not be the case. Only in appendix A.4 I could understand that they are regularly spaced in [0,1], and only in A.6.1 that the time is scaled between 0 and 1 in all datasets, "reference points" are mentioned in the "intensity encoding" and "model output" paragraphs. But for a reader not familiar with the mTAN it is not obvious why we are interested in them.*
>
> A: Thanks for pointing this out. We have added more details regarding this to the paper in the last paragraph of Section 3.2.
>
> *Q: The prior over z is not defined in the paper*
>
> A:  Each of these latent states are IID-distributed according to a standard multivariate normal distribution. We have added it to the paper.
>
> *Q: Learning output variances is normally not a problem in VAEs, why is it a problem in this case?*
>
> A: In general, a fixed output variance is used for training the VAE. Also, in most cases, people are interested in the mean output of the decoder. A VAE trained with a heteroscedastic output layer suffers from underfitting of predictive variance as evidenced by our work as well as [1, 2]. Actually, it is well known that naively attempting to learn the variance in a Gaussian decoder leads to poor results [3, 4, 5].
>
> *Q: How robust is the choice of the scaling factor lambda across different datasets?*
>
> A: If the dataset is z-normalized, then lambda between 1.0 and 5.0 works well.
>
> *Q: Choosing equally spaced reference points means that if one uses as a test point a sample temporally close to the input data (or even a sample from the input data), the imputation of the model might be unnecessarily poor. Could this be improved?*
>
> A: The choice of reference points is flexible. In some cases, we may have a fixed set of such points. In other cases, the set of
> reference time points may need to depend on the input time series itself. The number of reference points is also flexible and is used as a hyperparameter that controls how far the reference points are. So, it is possible that if we use very few reference points the imputation can be poor, but it can be improved by increasing the number of reference points.
>
>
> *Q: As stated by the authors, the model is only able to provide a marginal distribution at each time step, which practically means that the sampled trajectories might look inconsistent (non-smooth). Exploring the usage of sequential latent variable models for this would be an interesting future research direction.*
>
> A: Thanks for providing inputs on this. As we discuss in the conclusion section, this is an interesting direction for future work.
>
>
> *Q: The baselines for HVAE RNN  and HVAE RNN-ODE are much worse than the forward imputation, which makes me question the implementation of the models.*
>
> A: We note that in the case of the VAE-based models with homoscedastic output (or constant variance at the output) we treat the output variance as a hyperparameter and optimize it for better log-likelihood scores. This improves the log-likelihood values but worsens the MAE/MSE scores. The MAE/MSE scores of these models can be improved by using a small fixed variance during training, but that produces worse log-likelihood values. We have mentioned this in Section 4, 3rd paragraph.
>
> References:
> 1. Garnelo, Marta, et al. "Neural processes." arXiv preprint arXiv:1807.01622 (2018).
> 2. Kim, Hyunjik, et al. "Attentive neural processes." arXiv preprint arXiv:1901.05761 (2019).
> 3. Rybkin, Oleh, Kostas Daniilidis, and Sergey Levine. "Simple and effective VAE training with calibrated decoders." ICML 2021.
> 4. Rezende, D. J. and Viola, F. Taming vaes. arXiv preprint arXiv:1810.00597, 2018.
> 5. Dai, B. and Wipf, D. Diagnosing and enhancing vae models. arXiv preprint arXiv:1903.05789, 2019.

---

### Official Review · Reviewer_i1vq · 2021-10-29

**Correctness:** 4
**Technical Novelty And Significance:** 3
**Empirical Novelty And Significance:** 3
**Recommendation:** 6
**Confidence:** 3

**Main Review:**

## Contribution

### Advantages

The tackled problem of probabilistic interpolation of time series is relevant and especially valuable as it has received less attention than the forecasting task, at least in the neural networks community. This is especially the case in the considered setting where observations are not synchronized between dimensions.

The proposed model is for the most part interesting and well motivated, justifying the additions that are made on top of the prior mTAN. These contributions are, to the best of my knowledge, novel. The so-called heteroscedasticity of the output variance answers a crucial issue of standard VAEs with constant output variance in this context of application - even though this has already been considered elsewhere. The intensity branch of the introduced network clearly addresses the shortcomings of prior state-of-the-art methods. Accordingly, the paper is mostly well written and easy to read.

Experimentally, the performance of HeTVAE is state-of-the-art by a large margin, showing the benefits of the approach. Ablation studies show the individual contribution of each model component; I suggest that the authors include the full ablation of the appendix in the main paper. The experiments are sufficiently well designed and diverse on real-world applications in order to correctly assess this performance. Furthermore, qualitative experiments with examples of interpolation are appealing and highlight the impact of the model and its different components.

For all these reasons, I think that this paper is interesting and might be ready for publication at ICLR after the revision during the rebuttal. Indeed, I would express reservations, that I detail below.

### Limitations and Potential Improvements

A first limitation of the current version of the paper deals with the significance of the introduced heteroscedasticity. While very few models that are able tackle the same task as HeTVAE seem to leverage temporally varying output variance, it is unclear whether this is an inherent advantage of the described model, or if it is orthogonal to the other architectural considerations. In other words, could heteroscedasticity be reasonably applied to the considered RNN/ODE-based baselines? If it does, I recommend that the authors include these augmented baselines in their experiments to better contextualize the performance of HeTVAE. In any case, a more detailed discussion of the related work is necessary in this regard. In particular, I would advise the authors to consider other works which, to my understanding, could be included in the related work and considered baselines with heteroscedasticity [1, 2].

A second limitation is a partial lack of specification or explication of the modeling choices. In particular, the motivation and intuition behind the deterministic path is missing from the paper, to my understanding; I am wondering about the impact of its introduction in parallel of the usual variational latent variable, since the deterministic variable is unconstrained whereas the stochastic one is constrained by the KL term in the loss. An additional ablation showing the performance of HeTVAE without the stochastic branch would thus be interesting. Moreover, the role of the reference timestamps $\mathbf{r}$ is unclear and they are only specified in the appendix; could the authors discuss their necessity in the model and the relevance of their choice? Further discussion about how the baselines are adapted for the considered task should also be included. Finally, it is unclear until Section 4 that the learned VAE also learns to predict interpolations besides reconstructing its inputs: this information should be clearly stated earlier, probably in the description of the ELBO.

## Other Remarks and Questions

### Scalability of the Intensity Pathway

Could the authors comment on the scalability of the implementation of the intensity pathway as described in Equation (1)? To my understanding, the numerator pools over all elements of the dataset and it would seem that a large-scale dataset would prevent an efficient computation of this branch.

### Nature of the Augmented Learning Objective

There seems to be a typo in the Equation (11): the augmented learning objective should be minimized, so it should be negatively added the the ELBO.

### Code and Supplementary Material

Based on my limited review, the experiments of this paper seem to be reproducible. The provided code is appreciated. I recommend the authors to remove the hidden folders in the archive, especially the `.git` which is irrelevant and makes the archive heavier than needed.


## References

[1] X. Li et al. Scalable Gradients for Stochastic Differential Equations. AISTATS 2020.\
[2] A. Norcliffe et al. Neural ODE Processes. ICLR 2020.

**Summary Of The Paper:**

This paper introduces a novel model, HeTVAE, for probabilistic interpolation of time series that are irregularly sampled. HeTVAE builds on prior work by complementing it with a learned time-dependent output variance in the VAE and architectural improvements. The latter include a new branch accounting for the distribution of the sampled timestamps in the series and the addition of a deterministic branch bypassing the stochastic variational latent variable. The performance of HeTVAE is evaluated on multiple datasets against various baselines and via ablation studies.

**Summary Of The Review:**

Given the relevance and experimental results of the proposed model, I think that this paper may be valuable to the audience of ICLR. Nonetheless, some substantial limitations about the nature of the heteroscedasticity contribution and unclear points in the paper prevent me from giving a firm positive score, thus choosing a score marginally below the acceptance threshold.

I believe that this paper may be improved in the course of the discussion phase and would be pleased to raise my score, should my concerns be addressed during this time.

### Post-Rebuttal Update

I acknowledge the authors' response and thank them for their detailed answer. As explained in my follow-up response, the rebuttal partly addressed my concerns. While I still find that it could be improved on some aspects, I believe that this paper may be accepted to the conference. Therefore, before discussing with the other reviewers, I choose to raise my score from 5 to 6.

---

> ### Author Response · Authors · 2021-11-18
> **Response to Reviewer i1vq (Part 1)**
>
> We thank the reviewer for acknowledging the novelty and contributions and for providing a detailed and thorough review. We address the limitation and discuss the potential improvements below.
>
>
> **Q: A first limitation of the current version of the paper deals with the significance of the introduced heteroscedasticity. While very few models that are able tackle the same task as HeTVAE seem to leverage temporally varying output variance, it is unclear whether this is an inherent advantage of the described model, or if it is orthogonal to the other architectural considerations. In other words, could heteroscedasticity be reasonably applied to the considered RNN/ODE-based baselines? If it does, I recommend that the authors include these augmented baselines in their experiments to better contextualize the performance of HeTVAE. In any case, a more detailed discussion of the related work is necessary in this regard. In particular, I would advise the authors to consider other works which, to my understanding, could be included in the related work and considered baselines with heteroscedasticity [1, 2].**
>
> A: Our paper provides contributions from two angles. Our first contribution is the introduction of the intensity network that can directly encode information about input uncertainty due to variable sparsity. This is important in representing the variable output uncertainty at the output. We show through several experiments and ablations in Appendix A.2 and A.3.3 that without this component it is hard for the model to output variable uncertainty in sparse time series. This contribution is particular to the proposed model and cannot be applied to other RNN/ODE baselines.  Our second contribution is the introduction of an augmented learning objective to address the learning of VAE in the presence of a heteroscedastic output layer. As we have shown through several ablations (Appendix A.2 and A.3.1) and through several prior works [1,2] that naively attempting to learn the variance in a Gaussian decoder suffers from underfitting of predictive variance and leads to poor results [3, 4, 5]. We agree that this contribution is orthogonal to the other architectural contribution. We already provide ablation of the mTAN [6] to show the effect of the heteroscedastic layer. We chose mTAN for this experiment because when trained with constant output variance, mTAN consistently outperforms ODE and RNN-based baselines over all the metrics across all three datasets as shown in Tables 1, 2 and 3. Below are the results on two datasets:
>
> Results on PhysioNet dataset:
>
> | **Model**     |  **Negative Log-Likelihood** | Mean Absolute Error |	Mean Squared Error |
> | ----------- | ----------- | ----------- | ----------- |
> | mTAN (heteroscedastic layer)	 |	$	0.7866	\pm	0.0029	$	|	$	0.4857	\pm	0.0003	$	|	$	0.7120	\pm	0.0007	$ |
> | mTAN (constant variance) |	$	1.2426	\pm	0.0028	$	|	$	0.5056	\pm	0.0004	$	|	$	0.7167	\pm	0.0016$|
>
> Results on MIMIC-III dataset:
>
> | **Model**     |  **Negative Log-Likelihood** | Mean Absolute Error |	Mean Squared Error |
> | ----------- | ----------- | ----------- | ----------- |
> | mTAN (heteroscedastic layer)  |	$	0.9245	\pm	0.0021	$	|	$	0.5208	\pm	0.0009	$	|	$	0.5358	\pm	0.0012	$|
> | mTAN (constant variance) |	$	1.0498	\pm	0.0013	$	|	$	0.4931	\pm	0.0008	$	|$	0.4848	\pm	0.0008	$|
>
> As we can see that the introduction of the heteroscedastic layer certainly improves the log-likelihood scores on both datasets. For detailed ablation of several components of our model, please see appendix A.2. We would like to thank the reviewer for pointing out the related works. We have added a discussion in the related work.
>
> **Q: Moreover, the role of the reference timestamps  is unclear and they are only specified in the appendix; could the authors discuss their necessity in the model and the relevance of their choice?**
>
> A:  The UnTAN module defines a continuous function of $t$ given an input time series and hence cannot be directly incorporated into standard neural network architectures. We adapt the UnTAN module to produce fully observed fixed-dimensional discrete sequences by materializing its output at a set of reference time points. Reference time points can be fixed set of regularly spaced time points or may need to depend on the input time series. For the case of sparse and irregularly sampled time series which the datasets exhibit here, we chose regularly spaced points as reference time points. We treat the number of reference time points (which decides the granularity of representation) as a hyperparameter. We have added this in the last paragraph of Section 3.2.
>
> **Q: Finally, it is unclear until Section 4 that the learned VAE also learns to predict interpolations besides reconstructing its inputs: this information should be clearly stated earlier, probably in the description of the ELBO.**
>
> A: We have added add this information in Section 3 with the description of the ELBO.

---

> > ### Author Response · Authors · 2021-11-18
> > **Response to Reviewer i1vq (Part 2)**
> >
> > **Q: A second limitation is a partial lack of specification or explication of the modeling choices. In particular, the motivation and intuition behind the deterministic path is missing from the paper, to my understanding; I am wondering about the impact of its introduction in parallel of the usual variational latent variable, since the deterministic variable is unconstrained whereas the stochastic one is constrained by the KL term in the loss. An additional ablation showing the performance of HeTVAE without the stochastic branch would thus be interesting.**
> >
> > A: The inclusion of the deterministic pathway is empirically motivated. We have performed an additional ablation to show the performance gain obtained by the determinstic pathway in Appendix A.2. We include the results below:
> >
> > Results on PhysioNet dataset:
> >
> > | **Model**     |  **Negative Log-Likelihood** | Mean Absolute Error |	Mean Squared Error |
> > | ----------- | ----------- | ----------- | ----------- |
> > | HeTVAE	 | $	0.5542	\pm	0.0209 $ |	$ 0.3911	\pm	0.0004 $	|	$ 0.5778	\pm	0.0020 $ |
> > | HeTVAE - DET|	$	0.6278	\pm	0.0017	$	|	$	0.4089	\pm	0.0005	$	|	$	0.595	\pm	0.0018	$|
> > | HeTVAE - PROB|  $	0.7749	\pm	0.0047	$	|	$	0.4251	\pm	0.0029	$	|	$	0.623	\pm	0.004	$|
> >
> > Results on MIMIC-III dataset:
> >
> > | **Model**     |  **Negative Log-Likelihood** | Mean Absolute Error |	Mean Squared Error |
> > | ----------- | ----------- | ----------- | ----------- |
> > | HeTVAE |	$	0.6662	\pm	0.0023	$	|	$	0.3978	\pm	0.0003	$	|	$	0.3716	\pm	0.0001 $|
> > | HeTVAE - DET |	$	0.7478	\pm	0.0028	$	|	$	0.4129	\pm	0.0008	$	|	$	0.3845	\pm	0.0009	$|
> > |HeTVAE - PROB|  $	0.7472	\pm	0.0056	$	|	$	0.4049	\pm	0.0006	$	|	$	0.3833	\pm	0.0008	$|
> >
> > We can see that removing the deterministic pathway from the model results in a performance drop on both MIMIC-III and PhysioNet across all metrics.  This could be because the deterministic pathway has the ability to propagate the input sparsity information directly to the decoder (by skipping over the sampling), which is important for computing the variable output uncertainty.  We have also added an ablation showing the performance of HeTVAE without the stochastic branch or probabilistic pathway (HeTVAE - PROB). Removing the probabilistic pathway turns the model into an autoencoder with a heteroscedastic output layer.
> >
> >
> > **Q: Could the authors comment on the scalability of the implementation of the intensity pathway as described in Equation (1)? To my understanding, the numerator pools over all elements of the dataset and it would seem that a large-scale dataset would prevent an efficient computation of this branch.**
> >
> > A: Actually the denominator is computed over the whole dataset. We need to compute the union of all time points over the dataset just once during training and use them in every iteration. Hence, its scalable to larger datasets.
> >
> > **Q: There seems to be a typo in the Equation (11): the augmented learning objective should be minimized, so it should be negatively added the the ELBO.**
> >
> > A: Thanks for catching this, we have fixed this in the new version.
> >
> > **Based on my limited review, the experiments of this paper seem to be reproducible. The provided code is appreciated. I recommend the authors to remove the hidden folders in the archive, especially the .git which is irrelevant and makes the archive heavier than needed.**
> >
> > A: We'll make sure to remove the .git file.
> >
> >
> > References:
> >
> > 1. Garnelo, Marta, et al. "Neural processes." arXiv preprint arXiv:1807.01622 (2018).
> > 2. Kim, Hyunjik, et al. "Attentive neural processes." arXiv preprint arXiv:1901.05761 (2019).
> > 3. Rybkin, Oleh, Kostas Daniilidis, and Sergey Levine. "Simple and effective VAE training with calibrated decoders." ICML 2021.
> > 4. Rezende, D. J. and Viola, F. Taming vaes. arXiv preprint arXiv:1810.00597, 2018.
> > 5. Dai, B. and Wipf, D. Diagnosing and enhancing vae models. arXiv preprint arXiv:1903.05789, 2019.

---

> ### Comment · Reviewer_i1vq · 2021-11-23
> **Updated Review**
>
> ## Updated Score
>
> I would like to thank the authors for their detailed answer, which partly addressed my concerns. In particular, I appreciate the efforts that have been made to clarify the contributions and the details of the proposed architecture regarding reference timestamps and the deterministic branch. While I am unsure of the overall significance of the paper, it seems to me that the introduced architecture and described experiments might constitute an interesting novel contribution for the community, especially given the few papers tackling this topic. Therefore, I am now leaning towards acceptance.
>
> However, I encourage the authors to keep on improving the paper, especially concerning comparisons with other methods in the literature on the heteroscedasticity aspect of the contribution, either in the related work section or in the experiments. Indeed, it appears that heteroscedasticity for time series has already been considered by some works in the literature and their inapplicability to the setting of the paper should be further explicated. Nevertheless, I do not see this issue as a ground for rejection given the other merits of the paper.
>
> As a consequence, I choose to raise my score of "Empirical Novelty And Significance" from 2 to 3 and my overall score from 5 to 6.
>
> ## Scalability of the Intensity Pathway
>
> I indeed meant to reference the denominator instead of the numerator in my remark. Following the authors' response, I still do not understand how this is scalable to large datasets. The union of all timestamps over the dataset may be computed only once, but the pooling operation over all these timestamps (thus, over all training points as well) has to be performed at each iteration, which could lead to a high computational cost when the dataset is large. A discussion or a clarification of this point in the final version of the paper would be welcome.

---

> > ### Author Response · Authors · 2021-11-30
> > **Thanks for updating the score!**
> >
> > We would like to thank the reviewer for increasing the score. Regarding the intensity pathway, we would like to point out that the denominator computation is independent of the examples (both the reference points and union of all points are known at the start) and we need to compute this once for each iteration. So the denominator computation is dot product of a matrix of sizes $K\times d_e$ (query embedding with $K$ reference points) and $|t_u| \times d_e$ (union of all time points $t_u$). We also note that different sets could be used for $t_u$ including a fixed set of reference time points. This denominator computation acts as a normalizer for the intensity encoding. We have discussed this in Section 3.2 under intensity encoding description. We can add more details in the final version.

---

### Official Review · Reviewer_oEK3 · 2021-11-02

**Correctness:** 4
**Technical Novelty And Significance:** 3
**Empirical Novelty And Significance:** 3
**Recommendation:** 6
**Confidence:** 3

**Main Review:**

+
Novel method
Well written
Notation is clear and organized
Well evaluated on multiple datasets
Extensive ablation tests and discussion about the model components
Code available at review

-
Big novelty overlap with [1]

[1] Satya Narayan Shukla and Benjamin Marlin. Multi-time attention networks for irregularly sampled time series. In International Conference on Learning Representations, 2021


**Summary Of The Paper:**

The paper introduces a novel model of Variational Autoencoder that deals with irregularly sampled time series with a probabilistic approach to do time series interpolation.

The main contribution is the architecture by itself, its components, and the training process.
The model was evaluated on both real-world data sets from the medical and climate domain and synthetic data.


**Summary Of The Review:**

Very good paper with correct methodology. relevant results and a concerning overlap with a recent ICLR paper

---

> ### Author Response · Authors · 2021-11-15
> **Addressing the novelty gap**
>
> We thank the reviewer for providing a positive review. Before addressing the novelty overlap, we want to point out that the other reviewers think that the proposed model is "novel" (R1, R2), "interesting" (R1), "well-motivated" (R1, R2), "intensity encoding is novel" (R3), and "outperforms competing methods by a large margin" (R1, R2).
>
> (We abbreviate the reviewer i1vq, EXpk, 7Ard to R1, R2, R3, respectively.)
>
> The novel contributions of the proposed work are as follows:
>
> **Intensity Encoding in UnTAN layer:** We introduce a novel Intensity Encoding layer that encodes information about input uncertainty due to variable sparsity. As we have described in the paper, the mTAN module (or Value Encoding) [1] on its own cannot represent information about input sparsity because of the normalization of the attention weights. In fact, mTAN module is completely invariant to an additive increase or decrease in all of the attention weights. All the information about input uncertainty is represented by the attention weights and we lose this information after normalization. To address this concern, we introduce the UnTAN layer which consists of a novel Intensity Encoding layer that specifically focuses on representing information about the sparsity of observations. As can be seen from ablation experiments as well as sample interpolations, this is indeed true that without the Intensity Encoding layer the model is not able to effectively capture the input uncertainty.
>
> **Heteroscedastic layer and Augmented Learning Objective:** Another major difference with the mTAN framework [1] is that the proposed model produces a heteroscedastic output distribution. On the contrary, the mTAN framework like many VAEs uses a homoscedastic output distribution that assumes constant uncertainty. This means that the model can only reflect uncertainty through the variations in the VAE latent state. As we show through several experiments and ablations, this mechanism is insufficient to produce variable uncertainty over time. Since our aim is to reflect uncertainty due to input sparsity in the output distribution, the proposed model uses heteroscedastic output distribution to reflect variable uncertainty across time. This also allows the model for much more flexible uncertainty modeling at the output because now the framework can model uncertainty through the variations in the VAE latent state as well as through the variations in the predicted variance across time. However, the use of heteroscedastic output representation can lead the model to get stuck in poor local optima where the mean is essentially flat and all of the structure in the data is explained as noise. This happens because now the model is also predicting variance, in addition, to the mean and the model has the flexibility to explain all the structures in the data only through predicted variance. We propose an augmented learning objective to combat the presence of additional local optima that arise from the use of the heteroscedastic output structure. We verify this claim through several experiments and sample interpolations showing that the use of this augmented training objective has a strong positive impact on final model performance.
>
> **SOTA performance:** The proposed model significantly improves uncertainty quantification in the output interpolations compared to several baselines and state-of-the-art methods. On the other hand, the mTAN framework [1] and other VAE based baselines are not able to reflect the input uncertainty in the output interpolations. For example, the mTAN framework always shows approximately constant uncertainty in the output interpolations (because of the use of homoscedastic output distribution) (Figure 2) and produces over-confident probabilistic interpolations over large gaps (Appendix A.3.3 and Figure 5).
>
> Overall, the proposed model provides novel and significant capabilities over the mTAN framework in terms of much more flexible and accurate uncertainty modeling at the output.
>
> References:
> 1. Satya Narayan Shukla and Benjamin Marlin. Multi-time attention networks for irregularly sampled time series. In International Conference on Learning Representations, 2021

---

### Author Response · Authors · 2021-11-22
**Summary of the updates**

Thanks again to all of the reviewers for your comments on the paper. We have updated the paper based on the comments and discussions with the reviewers. Below are the list of updates:

- Added a discussion of related work [1,2,3] based on the recommendation of Reviewer  i1vq and 7Ard
- Added more details on the role of reference points and how are they selected based on the recommendation of Reviewer EXpk and i1vq
- Added a missing detail on defining prior over latent variables and fixed a small bug in the loss function as pointed out by Reviewer i1vq
- Added more details regarding the training of the proposed model based on the recommendation of Reviewer i1vq


References:
1. Siyuan Shan and Junier B. Oliva. NRTSI: Non-Recurrent Time Series Imputation for Irregularlysampled Data. ArXiv, abs/2102.03340, 2021.
2. Xuechen Li, Ricky Tian Qi Chen, Ting-Kam Leonard Wong, and David Duvenaud. Scalable gradients for stochastic differential equations. In Artificial Intelligence and Statistics, 2020.
3. Alexander Norcliffe, Cristian Bodnar, Ben Day, Jacob Moss, and Pietro Lio. Neural ` {ode} processes. In International Conference on Learning Representations, 2021.

---

### Decision · Program_Chairs · 2022-01-20

**Decision:**

Accept (Poster)

**Comment:**

The submission proposes a model to handle uncertainties in an irregularly sampling time series setting (HetVAE), built on the VAE framework and the previous work on mTAN (multi-time attention networks), and introduces components to encode sparsity information and heteroscedastic output uncertainty. The paper is clear, well-motivated, and contains extensive ablation studies showing the effect of eaach added components.
I recommend this submission for acceptance.